# Indirect Gradient Matching for Adversarial Robust Distillation

**Hongsin Lee\*, Seungju Cho\*, Changick Kim**
Korea Advanced Institute of Science and Technology (KAIST)
{hongsin04, joyga, changick}@kaist.ac.kr

## Abstract

Adversarial training significantly improves adversarial robustness, but superior performance is primarily attained with large models. This substantial performance gap for smaller models has spurred active research into adversarial distillation (AD) to mitigate the difference. Existing AD methods leverage the teacher's logits as a guide. In contrast to these approaches, we aim to transfer another piece of knowledge from the teacher, the input gradient. In this paper, we propose a distillation module termed Indirect Gradient Distillation Module (IGDM) that indirectly matches the student's input gradient with that of the teacher. Experimental results show that IGDM seamlessly integrates with existing AD methods, significantly enhancing their performance. Particularly, utilizing IGDM on the CIFAR-100 dataset improves the AutoAttack accuracy from 28.06% to 30.32% with the ResNet-18 architecture and from 26.18% to 29.32% with the MobileNetV2 architecture when integrated into the SOTA method without additional data augmentation.

## 1 Introduction

Recently, adversarial attacks have revealed the vulnerabilities of deep learning-based models (Goodfellow et al., 2014; Carlini & Wagner, 2017; Madry et al., 2017), raising critical concerns in safety-important applications (Grigorescu et al., 2020; Ma et al., 2021; Wang et al., 2023a). Thus, much research has been done on defense technologies to make deep learning more reliable (Das et al., 2017; Carmon et al., 2019; Cohen et al., 2019; Xie et al., 2019; Zhang et al., 2022; Jin et al., 2023). Among adversarial defense mechanisms, adversarial training is one of the most effective methods to enhance adversarial robustness (Pang et al., 2020; Bai et al., 2021a; Wei et al., 2023). However, there is a significant performance gap between large and small models in adversarial training. Since light models with less computational complexity are preferred in practical applications, increasing the robustness of light models is necessary. To address this, adversarial training incorporates distillation methodologies which are commonly employed to boost the performance of light or small models (Goldblum et al., 2020; Zhu et al., 2021; Zi et al., 2021; Huang et al., 2023).

In the teacher-student architecture of knowledge distillation, prevailing methods leverage the teacher's features or logits (Hinton et al., 2015; Ji et al., 2021; Kim et al., 2021; Yang et al., 2022). Adversarial distillation (AD) approaches extend this paradigm by incorporating the teacher's logits as a guide within their adversarial training framework (Goldblum et al., 2020; Zhu et al., 2021; Maroto et al., 2022). Recent studies have specifically focused on tailoring the inner maximization problem of adversarial training, particularly by involving the teacher model in the generation of adversarial examples (Zi et al., 2021; Huang et al., 2023). We leverage input gradients, which have been studied in relation to robustness in adversarially trained models (Tsipras et al., 2018; Engstrom et al., 2019; Srinivas et al., 2023).

We additionally demonstrate that matching the input gradients between teacher and student contributes to point-wise alignment, which has been studied in state-of-the-art AD methods (Huang et al., 2023). In the top right of Figure 1a, if the input gradients between the teacher and the student match as depicted in the red line, then the output of the points located on the red line, denoted as

---

*These authors contributed equally.

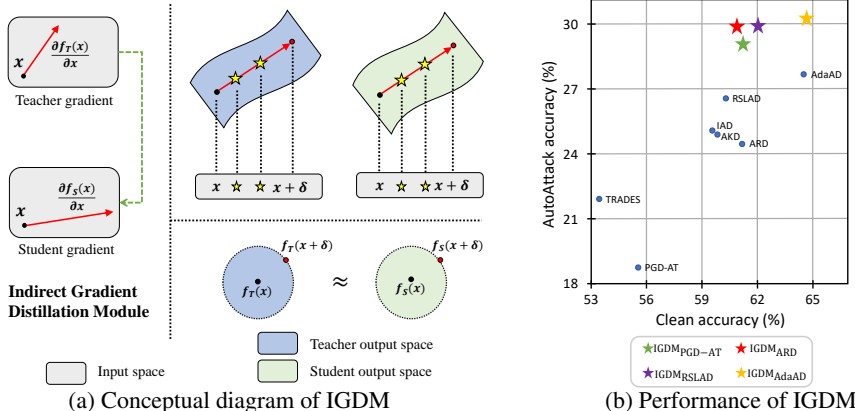

(a) Conceptual diagram of IGDM      (b) Performance of IGDM

Figure 1: **(a)** We match the gradient in the input space with knowledge distillation. Since adversarial robust models are locally linear, matching the gradient has the effect of matching the output points of the surrounding input marked with 'stars', as illustrated in the top right figure. Consequently, gradient matching involves matching the teacher and student point by point in the output space, as depicted in the bottom right. **(b)** Clean and Autoattack accuracy on ResNet-18 with BDM-AT teacher on CIFAR-100 dataset. IGDM demonstrates significantly improved AutoAttack accuracy compared with other adversarial distillation methods.

*stars*, will also match. We show that the output of random points around $x$ also becomes similar to the teacher as illustrated in the bottom right of Figure 1a. This alignment causes the student to mimic the teacher and reduces the capacity gap between teacher and student.

In this paper, we propose *Indirect Gradient Distillation Module* (IGDM), which indirectly aligns the student's gradient with that of the teacher. We can easily match input gradients through Taylor expansion, leveraging the locally-linear property of adversarial training. Since existing AD methods are mainly designed to match logits, IGDM can be used in conjunction with them. Through extensive experimental results, we verify that our method successfully complements the robustness of existing AD methods. We significantly improve existing AD methods as depicted in Figure 1b. Our contributions are as follows:

- We propose a methodology to transfer the gradient information from the teacher to the student through the exploitation of the local linearity inherent in adversarial training. This stands in contrast to prevailing AD methods, which primarily concentrate on the distillation of logits.
- Based on the analysis, we propose the Indirect Gradient Distillation Module (IGDM), which indirectly distills the gradient information. Its modular design allows easy integration into existing AD methods.
- We empirically demonstrate that IGDM significantly improves robustness against various attack scenarios, datasets, student models, and teacher models.

## 2    RELATED WORK

### 2.1    ADVERSARIAL TRAINING

Adversarial training (AT) involves generating adversarial examples during the training process to improve model robustness. The adversarial training is a minimization-maximization problem as follows:

$$\arg\min_{\theta} \mathbb{E}_{(\mathbf{x},y)\sim D}\left[L_{min}\left(f_\theta(\mathbf{x}+\boldsymbol{\delta}),y\right)\right], \text{ where } \boldsymbol{\delta} = \arg\max_{\|\boldsymbol{\delta'}\|_p\leq\epsilon} L_{max}\left(f_\theta(\mathbf{x}+\boldsymbol{\delta'}),y\right). \quad (1)$$

Here, $\theta$ is parameters of model $f$, $D$ is a data distribution of $\mathbf{x}$ and corresponding labels $y$, $\boldsymbol{\delta}$ is perturbation causing the largest loss within $l_p$ norm of $\epsilon$, $L_{max}$ is an inner maximiazation loss, and $L_{min}$ is an outer minimization loss. Multi-step PGD attack (Madry et al., 2017) is commonly utilized to

solve the inner maximization loss, and various regularization loss functions have been introduced for the outer minimization loss. TRADES (Zhang et al., 2019) incorporates the Kullback-Leibler (KL) divergence loss between the predictions of clean and adversarial images. MART(Wang et al., 2020) introduces per-sample weights based on the confidence of each sample. Due to their simplicity and effectiveness, TRADES and MART are commonly used as baseline methods in adversarial training (Wu et al., 2020; Bai et al., 2021b; Jin et al., 2022; Tack et al., 2022; Jin et al., 2023; Wei et al., 2023; Qin et al., 2019). Moreover, several strategies such as data augmentation (Rebuffi et al., 2021; Li & Spratling, 2023), and diverse loss functions (Wu et al., 2020), have been introduced.

Although defending against strong adversarial attack strategies(Croce & Hein, 2020a;b; Andriushchenko et al., 2020) is challenging, highly robust models have also been developed. Low Temperature Distillation (LTD) (Chen & Lee, 2021) points out the shortcomings of one-hot labels in adversarial training and advocates the use of soft labels as an alternative approach. Better Diffusion Models for Adversarial Training (BDM-AT) (Wang et al., 2023b) explores methods for the more efficient utilization of diffusion models within the context of adversarial training. Improved Kullback–Leibler Adversarial Training (IKL-AT) (Cui et al., 2023) inspects the mechanism of KL divergence loss and proposes the Decoupled Kullback-Leibler divergence loss. However, given the use of large architectures in these models, there is a necessity to enhance adversarial robustness in smaller models.

## 2.2 ADVERSARIAL ROBUST DISTILLATION

Adversarial distillation (AD) is an effective technique to distill the robustness from large teacher model to small student model (Goldblum et al., 2020; Zhu et al., 2021; Maroto et al., 2022; Zi et al., 2021; Huang et al., 2023; Kuang et al., 2023; Jung et al., 2024; Yin et al., 2024; Dong et al., 2025). Adversarial Robustness Distillation (ARD) (Goldblum et al., 2020) reveals that students can more effectively acquire robustness when guided by a robust teacher within an adversarial training framework. RSLAD (Zi et al., 2021) emphasizes the importance of smooth teacher logits in robust distillation, integrating the teacher's guidance directly into the adversarial image generation process. Introspective Adversarial Distillation (IAD) (Zhu et al., 2021) concentrates on assessing the reliability of the teacher and introduces a confidence score of the information provided by the teacher. AdaAD (Huang et al., 2023) generates more sophisticated adversarial images through the integration of the teacher during the inner maximization process. Previous AD methods utilize the teacher's logits as a guide, but our approach also incorporates the distillation of gradient information.

## 2.3 GRADIENT DISTILLATION AND INPUT GRADIENT

In knowledge distillation, gradient information has been used in various ways (Czarnecki et al., 2017; Du et al., 2020; Zhu & Wang, 2021; Lan & Tian, 2023; Wang et al., 2022), computed in either input space or weight space. For example, an exploration of the diversity among teacher models in the weight gradient space aids in identifying an optimal direction for training the student network (Du et al., 2020). Another study examines the capacity gap between teachers and students, focusing on the perspective of weight gradients similarity (Zhu & Wang, 2021). Conversely, input space gradients find applications in knowledge distillation for tasks such as classification (Czarnecki et al., 2017), object detection (Lan & Tian, 2023), or language model (Wang et al., 2022). These methods all compute the gradient directly, which is different from our approach.

Additionally, input gradients of adversarially trained models contain semantically meaningful information, aligning with human perception, where image modifications for a specific class resemble human-recognized features (Tsipras et al., 2018; Engstrom et al., 2019; Srinivas et al., 2023). This suggests that robust models capture more human-aligned features, underlining the potential of distilling such valuable insights from robust teacher models to enhance student model performance in adversarial settings.

## 3 METHOD

In this section, we elaborate on how to distill the input gradients from the robust teacher models. In the analysis of Appendix A.4, direct gradient matching is shown to be ineffective. Therefore, we propose a novel approach for gradient matching that avoids direct gradient computation.

## 3.1 LOCAL LINEARITY OF ADVERSARIAL TRAINING

First, we show that adversarially trained models are capable of first-order Taylor expansion on the input unlike natural training. For small noise $\epsilon$, the output of $\mathbf{x} + \epsilon$ can be expressed as follows.

$$f(\mathbf{x}+\epsilon) = \underbrace{f(\mathbf{x}) + \left(\frac{\partial f(\mathbf{x})}{\partial \mathbf{x}}\right)^{T} \epsilon}_{\text{first-order approximation}} + \underbrace{O(\|\epsilon\|^2)}_{\text{remainder}}. \quad (2)$$

Here, $f$ denotes the model, where we use $f$ for brevity instead of $f_\theta$, and $\epsilon$ is sufficiently small noise with the same dimension as the $\mathbf{x}$. To explore the influence of the remainder, we calculate the proportion occupied by the remainder in Equation (2) by introducing uniform noise with a magnitude of $8/255$ as the perturbation $\epsilon$. We calculate the remainder proportion as the ratio of the remainder to the total value, *i.e.*, $\frac{\|f(\mathbf{x}+\epsilon)-f(\mathbf{x})-\left(\frac{\partial f(\mathbf{x})}{\partial \mathbf{x}}\right)^{T}\epsilon\|}{\|f(\mathbf{x}+\epsilon)\|}$. We first examine the remainder proportion in an adversarially well-trained model: LTD (Chen & Lee,

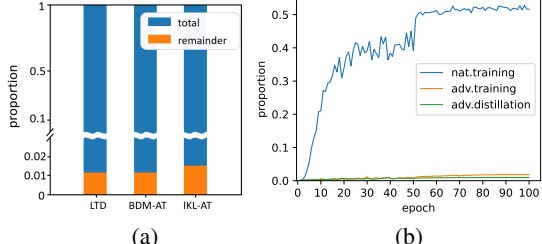

(a)            (b)

Figure 2: The proportion occupied by the remainder in Taylor expansion. **(a)** The remainder proportion of an adversarially robust teacher model on CIFAR-100 dataset.

**(b)** The proportion along with training epochs in natural training, adversarial training, and adversarial distillation using ResNet-18 on CIFAR-100 dataset.

2021), BDM-AT (Wang et al., 2023b), and IKL-AT (Cui et al., 2023) where we summarized the performance in Table 1. In Figure 2a, the remainder proportions are computed to be very small, with values of 0.012, 0.012, and 0.016 for three adversarially-trained models LTD, BDM-AT, and IKL-AT, respectively. In other words, $f(\mathbf{x} + \epsilon)$ can be approximated to the first-order Taylor expansion since the remainder proportion is negligible. Thus, we utilize this local linearity of the adversarially well-trained models to match input gradients.

Next, we investigate whether the model retains the local linearity in adversarial training, not in the case of a fully-trained model. We test three training strategies: natural training, adversarial training via ten steps of PGD (Madry et al., 2017), and adversarial distillation (Goldblum et al., 2020) using the BDM-AT teacher. In Figure 2b, the remainder proportion continuously increases in natural training, while it consistently remains small in both adversarial training and adversarial distillation. Therefore, employing first-order Taylor expansion is feasible during adversarial training. In the following section, we demonstrate how the locally linear property enables the matching of input gradients between the teacher and the student.

## 3.2 GRADIENT MATCHING VIA OUTPUT DIFFERENCES

For an input $\mathbf{x}$, we define $\mathbf{x}_{\epsilon_1}$ and $\mathbf{x}_{\epsilon_2}$ as follows:

$$\mathbf{x}_{\epsilon_1} = \mathbf{x} + \epsilon_1, \ \ \mathbf{x}_{\epsilon_2} = \mathbf{x} + \epsilon_2, \quad (3)$$

where $\epsilon_1$ and $\epsilon_2$ represent small random perturbations of the same dimension as the input $\mathbf{x}$. For adversarially trained or training model $f$, the output of $\mathbf{x}_{\epsilon_1}$ and $\mathbf{x}_{\epsilon_2}$ can be approximated using first-order Taylor expansion through the input space:

$$f(\mathbf{x}_{\epsilon_1}) \approx f(\mathbf{x}) + \left(\frac{\partial f(\mathbf{x})}{\partial \mathbf{x}}\right)^{T} \epsilon_1, \ \ f(\mathbf{x}_{\epsilon_2}) \approx f(\mathbf{x}) + \left(\frac{\partial f(\mathbf{x})}{\partial \mathbf{x}}\right)^{T} \epsilon_2, \quad (4)$$

where we neglect the remainder term based on the observations in Section 3.1. To extract and align the gradients between the student and the teacher models, we utilize the output differences as follows:

$$L = D\left(f_S(\mathbf{x}_{\epsilon_1}) - f_S(\mathbf{x}_{\epsilon_2}), \ f_T(\mathbf{x}_{\epsilon_1}) - f_T(\mathbf{x}_{\epsilon_2})\right), \quad (5)$$

where $f_S$ and $f_T$ represent the student and teacher models, while $D$ denotes a discrepancy metric like L2 or KL divergence loss. This loss can be reformulated using Equation (4) as follows:

$$L = D\left(\left(\frac{\partial f_S(\mathbf{x})}{\partial \mathbf{x}}\right)^{T} (\epsilon_1 - \epsilon_2), \ \left(\frac{\partial f_T(\mathbf{x})}{\partial \mathbf{x}}\right)^{T} (\epsilon_1 - \epsilon_2)\right). \quad (6)$$

If we consider $\epsilon_1 - \epsilon_2$ is an arbitrary perturbation vector, varying through the optimization process, minimizing this loss encourages the alignment of the gradients between the student and the teacher models, meaning

$$\frac{\partial f_S(\mathbf{x})}{\partial \mathbf{x}} \approx \frac{\partial f_T(\mathbf{x})}{\partial \mathbf{x}}. \tag{7}$$

### 3.3 INDIRECT GRADIENT DISTILLATION MODULE (IGDM)

To effectively integrate the gradient matching through output differences into AD methods, we select the $\epsilon_1$ and $\epsilon_2$ as constant multiples of the adversary perturbation $\delta$ from the AD methods, *i.e.*, $\epsilon_1 - \epsilon_2 \propto \delta$. Since the adversarial perturbations obtained during the training process continuously change, the gradients can be aligned. Specifically, as the adversarial perturbation $\delta$ employs gradient information, the inner product of gradient and $\epsilon_1 - \epsilon_2$ in Equation (6) can be effectively increased with the adversarial perturbation. In other words, $(\frac{\partial f_S(\mathbf{x})}{\partial \mathbf{x}})^T (\epsilon_1 - \epsilon_2) \propto (\frac{\partial f_S(\mathbf{x})}{\partial \mathbf{x}})^T \delta$ remains large, facilitating the enhanced alignment of gradients using loss function in Equation (6) during training.

Finally, Indirect Gradient Distillation Module (IGDM) loss is formulated as:

$$L_{IGDM} = D\left(f_S(\mathbf{x} + \boldsymbol{\delta}) - f_S(\mathbf{x} - \boldsymbol{\delta}),\ f_T(\mathbf{x} + \boldsymbol{\delta}) - f_T(\mathbf{x} - \boldsymbol{\delta})\right). \tag{8}$$

Since IGDM complements the loss function by matching the gradients between the teacher and student models, the integration of IGDM with other AD methods becomes feasible in the following manner:

$$L_{min} = L_{AD} + \alpha \cdot L_{IGDM}, \tag{9}$$

where $\alpha$ is a hyperparameter and $L_{AD}$ stands for the outer minimization loss of other AD methods such as ARD (Goldblum et al., 2020), RSLAD (Zi et al., 2021), AdaAD (Huang et al., 2023), etc.

## 4 EXPERIMENTS

We explain the experimental setup, followed by comparative performance evaluations of the proposed IGDM method against various AT and AD methods.

### 4.1 SETTINGS

**Teacher and Student Models**  We selected three teacher models including LTD (Chen & Lee, 2021), BDM-AT (Wang et al., 2023b), and IKL-AT (Cui et al., 2023) for CIFAR-10/100. The LTD model is widely adopted in prior AD research, while the others have achieved high-ranking performance in RobustBench (Croce et al., 2021), demonstrating superior robustness against AutoAttack (Croce & Hein, 2020a). For Tiny-ImageNet, we employed a pre-trained RiFT model (Zhu et al., 2023), and for SVHN, we trained a WideResNet-34-10 (Zagoruyko & Komodakis, 2016) with PGD-AT. For student models, we employed the ResNet-18 (He et al., 2016a) and MobileNetV2 (Sandler et al., 2018) for CIFAR-10/100, ResNet-18 for SVHN, and PreActResNet-18 (He et al., 2016b) for Tiny-ImageNet.

Table 1: Performance (%) of the teacher models. Experiments with teacher models in *italics* are provided in the supplementary material.

| Dataset | Teacher name | Architecture | Clean | PGD | AA |
|---|---|---|---|---|---|
| CIFAR-100 | BDM-AT (Wang et al., 2023b) | WideResNet-28-10 | 72.58 | 44.24 | 38.83 |
|  | LTD (Chen & Lee, 2021) | WideResNet-34-10 | 64.07 | 36.61 | 30.57 |
|  | *IKL-AT (Cui et al., 2023)* | WideResNet-28-10 | 73.80 | 44.14 | 39.18 |
| CIFAR-10 | LTD (Chen & Lee, 2021) | WideResNet-34-10 | 85.21 | 60.89 | 56.94 |
|  | *BDM-AT (Wang et al., 2023b)* | WideResNet-28-10 | 92.44 | 70.63 | 67.31 |
|  | *IKL-AT (Cui et al., 2023)* | WideResNet-28-10 | 92.16 | 71.09 | 67.73 |
| SVHN | PGD-AT (Madry et al., 2017) | WideResNet-34-10 | 93.90 | 61.78 | 54.28 |
| Tiny-ImageNet | *RiFT (Zhu et al., 2023)* | WideResNet-34-10 | 52.54 | 25.52 | 21.78 |

Table 2: Adversarial distillation result on ResNet-18 with two teacher models on CIFAR-100. The Clean, PGD, FGSM, C&W, and AA each indicate performance (%). Bold indicates cases where IGDM improved accuracy by more than 0.5 percentage points.

| Method | CIFAR-100 with BDM-AT teacher | | | | | CIFAR-100 with LTD teacher | | | | |
|--------|-------|------|------|------|------|-------|------|------|------|------|
|        | Clean | FGSM | PGD  | C&W  | AA   | Clean | FGSM | PGD  | C&W  | AA   |
| PGD-AT | 55.80 | 23.51 | 19.88 | 20.46 | 18.86 | 55.80 | 23.51 | 19.88 | 20.46 | 18.86 |
| +IGDM  | **60.83** | **38.11** | **35.09** | **30.21** | **29.16** | **60.49** | 37.22 | **33.72** | **29.61** | **28.02** |
| TRADES | 53.56 | 29.85 | 25.85 | 23.32 | 22.02 | 53.56 | 29.85 | 25.85 | 23.92 | 22.02 |
| +IGDM  | **60.88** | **36.26** | **32.26** | **26.50** | **25.50** | **59.29** | 36.08 | **32.08** | **26.10** | **26.10** |
| ARD    | 61.51 | 34.23 | 30.23 | 26.97 | 24.77 | 61.34 | 35.19 | 31.19 | 27.74 | 25.74 |
| +IGDM  | 61.62 | **39.75** | **35.75** | **30.99** | **28.79** | 61.58 | **37.45** | **33.45** | **29.94** | **27.84** |
| IAD    | 59.92 | 35.47 | 31.47 | 26.91 | 25.15 | 60.12 | 36.91 | 32.91 | 28.29 | 26.29 |
| +IGDM  | **62.99** | **37.32** | **34.76** | **29.55** | **27.76** | **62.73** | **37.69** | **33.69** | **29.65** | **27.49** |
| AKD    | 60.27 | 35.38 | 31.38 | 26.29 | 25.09 | 60.46 | 35.53 | 31.53 | 27.29 | 25.37 |
| +IGDM  | 60.42 | **37.63** | **33.62** | **29.27** | **28.86** | 60.31 | **38.15** | **34.15** | **29.44** | **28.44** |
| RSLAD  | 60.22 | 36.16 | 32.16 | 27.96 | 26.76 | 60.01 | 36.39 | 32.39 | 28.94 | 26.94 |
| +IGDM  | **62.06** | **39.67** | **35.67** | **30.98** | **29.78** | 60.43 | **38.02** | **34.02** | **29.94** | **28.51** |
| AdaAD  | 64.43 | 37.33 | 33.21 | 29.53 | 28.06 | 63.34 | 37.39 | 33.39 | 29.73 | 27.81 |
| AdaIAD | 64.13 | 37.33 | 33.33 | 29.02 | 27.82 | 63.24 | 37.45 | 33.45 | 29.04 | 27.83 |
| +IGDM  | 64.44 | **39.31** | **36.19** | **31.75** | **30.32** | 63.44 | **38.23** | **34.23** | **31.09** | **28.87** |

Table 3: Adversarial distillation result on ResNet-18 on CIFAR-10 and SVHN. The Clean, PGD, FGSM, C&W, and AA each indicate performance (%). Bold indicates cases where IGDM improved accuracy by more than 0.5 percentage points.

| Method | CIFAR-10 with LTD teacher | | | | | SVHN with PGD-AT teacher | | | | |
|--------|-------|------|------|------|------|-------|------|------|------|------|
|        | Clean | FGSM | PGD  | C&W  | AA   | Clean | FGSM | PGD  | C&W  | AA   |
| PGD-AT | 84.52 | 52.42 | 42.80 | 42.98 | 41.12 | 91.62 | 65.93 | 48.07 | 48.34 | 42.46 |
| +IGDM  | 84.15 | **59.21** | **53.70** | **51.19** | **49.63** | **92.57** | **72.19** | **60.55** | **56.10** | **52.12** |
| TRADES | 82.46 | 56.97 | 49.13 | 47.98 | 47.09 | 89.91 | 69.81 | 57.52 | 51.32 | 50.74 |
| +IGDM  | **83.50** | **60.84** | **54.64** | **49.34** | **48.83** | **91.98** | **71.85** | **60.09** | **56.68** | **54.15** |
| ARD    | 85.04 | 60.31 | 53.27 | 50.27 | 49.49 | 92.32 | 70.46 | 55.62 | 53.07 | 48.24 |
| +IGDM  | 85.18 | **61.24** | **54.75** | **51.31** | **50.20** | 92.19 | **72.01** | **60.07** | **56.56** | **52.95** |
| IAD    | 84.33 | 61.21 | 54.24 | 50.97 | 50.09 | 91.62 | 70.82 | 56.42 | 53.01 | 47.76 |
| +IGDM  | 84.49 | **62.45** | **56.55** | **53.01** | **51.09** | **93.03** | **71.85** | **58.86** | **54.66** | **51.45** |
| AKD    | 85.10 | 59.07 | 51.53 | 49.13 | 48.04 | 92.49 | 70.44 | 56.49 | 53.91 | 50.17 |
| +IGDM  | 84.94 | **61.36** | **54.89** | **51.55** | **50.87** | 92.50 | **71.15** | **59.38** | **55.97** | **52.51** |
| RSLAD  | 83.59 | 60.97 | 55.98 | 53.15 | 52.13 | 90.52 | 62.74 | 53.80 | 50.01 | 48.41 |
| +IGDM  | 83.67 | **62.41** | **57.34** | **54.00** | **53.10** | 90.71 | **64.71** | **56.14** | **53.98** | **50.69** |
| AdaAD  | 84.74 | 61.87 | 56.78 | 53.51 | 52.79 | 93.27 | 67.14 | 57.43 | 54.88 | 52.93 |
| AdaIAD | 84.75 | 61.98 | 57.04 | 53.57 | 52.88 | 93.39 | 67.10 | 57.26 | 54.76 | 52.74 |
| +IGDM  | 84.83 | **62.54** | **57.61** | **55.09** | **54.02** | 93.48 | **68.34** | **58.74** | **56.03** | **53.89** |

**Evaluation Metrics** After training, we evaluate performance using five metrics: Clean, FGSM, PGD, C&W, and AutoAttack (AA) accuracy. Clean refers to the accuracy on the test dataset. We measure FGSM and PGD accuracy against fast gradient sign method (FGSM) (Goodfellow et al., 2014) and 20-step projected gradient descent (PGD) attacks (Madry et al., 2017), respectively. The C&W attack measures accuracy against Carlini & Wagner (2017), while AA evaluates accuracy against the AutoAttack method (Croce & Hein, 2020a). All attacks were conducted within an $l_\infty$-norm bound of $8/255$.

**Implementation** We utilized the CIFAR-10/100 (Krizhevsky et al., 2009), SVHN (Netzer et al., 2011), and Tiny-ImageNet (Le & Yang, 2015) datasets for our experiments. Random crop and random horizontal flip were applied, while other augmentations were not utilized. Our training methods encompassed conventional adversarial training methods, PGD-AT (Madry et al., 2017) and TRADES (Zhang et al., 2019), as well as adversarial distillation techniques including ARD (Goldblum et al., 2020), IAD (Zhu et al., 2021), AKD (Maroto et al., 2022), RSLAD (Zi et al., 2021), and AdaAD (Huang et al., 2023). In our comparative analysis, we integrated IGDM into all of these methods, and we used shortened notations to represent the methods with IGDM. For example, when IGDM is combined with ARD, we denote it as **ARD + IGDM** or IGDM$_{ARD}$. We employed the recommended inner loss functions for generating adversarial examples as outlined in each baseline as specified in Appendix B. We fixed the hyperparameters of all given methods used in the experiment as original paper. Then, we adjusted the hyperparameter $\alpha$ of IGDM. For the surrogate loss of IGDM, we employed KL divergence loss, but using alternative surrogate losses, such as L1 and L2 loss, yields nearly indistinguishable results. The detailed experimental setting can be found in Appendix B.

## 4.2 RESULTS

**Main Results** We present the comprehensive results of integrating IGDM into other baseline methods and their original versions in Table 2, Table 3. More experimental results are in the appendix: distillation results on MobileNetV2 student (Appendix A.1), distillation with IKL-AT teacher (Appendix A.2), and experiments on the Tiny-ImagNet dataset (Appendix A.3). IGDM significantly improves robustness against various attacks with consistent clean accuracy, regardless of the original methods, datasets, student models, or teacher models. IGDM demonstrates notably enhanced AA robustness on the CIFAR-100 dataset, achieving 30.32% on ResNet-18 with a BDM-AT teacher.

**Gradient Alignment** We justify that IGDM can align the student's input gradient with the teacher's input gradient. To quantify this alignment between the two input gradients, we introduce two metrics: mean Gradient Distance (GD) and mean Gradient Cosine similarity (GC). GD quantifies the average L2 distance between the input gradients of the teacher and student models for all test samples, with smaller values indicating closer alignment. GC measures the cosine similarity between these gradients, where a value closer to one signifies better alignment. In Table 4, IGDM improves gradient alignment with the teacher model, with improved robustness. As shown in Figure 3, the robustness of the student model increases as GD decreases and GC rises, confirming the positive correlation between gradient alignment and student model performance.

Table 4: Gradient alignment on CIFAR-100, using the ResNet-18 student model. The numbers in bold indicate enhanced gradient alignment.

| Method | BDM-AT | | | LTD | | |
|---|---|---|---|---|---|---|
| | AA | GD⇓ | GC⇑ | AA | GD⇓ | GC⇑ |
| ARD | 24.77 | 0.142 | 0.439 | 25.74 | 0.108 | 0.592 |
| **+IGDM** | **28.79** | **0.101** | **0.571** | **27.84** | **0.082** | **0.688** |
| IAD | 25.15 | 0.135 | 0.443 | 26.29 | 0.102 | 0.596 |
| **+IGDM** | **27.76** | **0.104** | **0.549** | **27.49** | **0.086** | **0.674** |
| AKD | 25.09 | 0.127 | 0.438 | 25.37 | 0.113 | 0.584 |
| **+IGDM** | **28.86** | **0.114** | **0.513** | **28.44** | **0.078** | **0.693** |
| RSLAD | 26.76 | 0.118 | 0.492 | 26.94 | 0.089 | 0.658 |
| **+IGDM** | **29.78** | **0.096** | **0.582** | **28.51** | **0.077** | **0.709** |
| AdaAD | 28.06 | 0.107 | 0.567 | 27.81 | 0.077 | 0.736 |
| AdaIAD | 27.82 | 0.107 | 0.568 | 27.83 | 0.077 | 0.733 |
| **+IGDM** | **30.32** | **0.086** | **0.643** | **28.87** | **0.070** | **0.769** |

**Point-wise Alignment** Existing adversarial distillation methods have largely overlooked the importance of input gradients, instead focusing on point-wise alignment. For example, RSLAD (Zi et al., 2021) and AdaAD (Huang et al., 2023) explicitly aim to align clean and adversarial outputs with the teacher's clean output. Thus, we reinterpret point-wise alignment in terms of input gradient matching, where IGDM demonstrates superior performance compared to existing AD methods in achieving this alignment. We define point-wise distance $D(\mathbf{x}, \boldsymbol{\delta}) = \|f_T(\mathbf{x}+\boldsymbol{\delta}) - f_S(\mathbf{x}+\boldsymbol{\delta})\|$. Then given $\mathbf{x}$, the upper bound of $D(\mathbf{x}, \boldsymbol{\delta})$ for sufficiently small $\boldsymbol{\delta}$ is as follows.

$$D(\mathbf{x}, \boldsymbol{\delta}) \leq \|(f_T(\mathbf{x}) - f_S(\mathbf{x})\| + \left\| \left( \frac{\partial f_T(\mathbf{x})}{\partial \mathbf{x}} - \frac{\partial f_S(\mathbf{x})}{\partial \mathbf{x}} \right)^T \boldsymbol{\delta} \right\|. \tag{10}$$

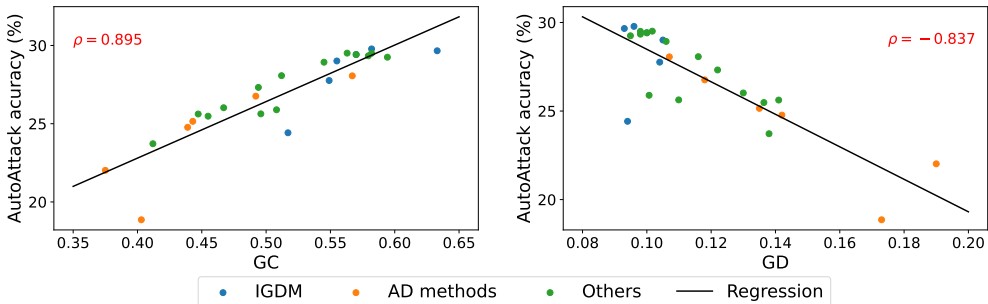

Figure 3: Correlation between GC and AA ($left$) and between GD and AA ($right$). All results were obtained using ResNet-18 and a BDM-AT teacher on CIFAR-100. The values for IGDM and AD methods match those in Table 2, while 'Others' represent results from additional experiments under the same configuration. $\rho$ denotes the correlation coefficient.

Hence, the better the gradient matching, the smaller the upper bound of $D(\mathbf{x}, \boldsymbol{\delta})$ becomes, aligning the teacher and student in a point-wise manner.

To empirically substantiate this assertion, we compared the distances between the teacher and the student models. We calculated $D(f_S(\mathbf{x}+\boldsymbol{\delta}), f_T(\mathbf{x}+\boldsymbol{\delta}))$ with L2 distance for random noise from the uniform distribution and adversarial perturbation in Table 5. For the adversarial noise, we conducted an adversarial attack using the inner maximization loss corresponding to each method, with a fixed number of steps set to 20. In all cases, we observed an improvement in alignment when using IGDM. In particular, IGDM enhanced alignment against not only adversarial noise but also random noise. This outcome demonstrates that our module contributes to point-wise alignment, improving overall robustness by significantly reducing local invariance during training, as asserted in Huang et al. (2023).

Table 5: Point-wise alignment with IGDM on CIFAR-100, using the ResNet-18 student model and two teacher models. The $\boldsymbol{\delta} \sim U$ indicates uniformly selected $\boldsymbol{\delta}$ from $U[-8/255, 8/255]$. The numbers in bold indicate enhanced point alignment.

| Method | BDM-AT Teacher | | LTD teacher | | IKL-AT teacher | |
|---|---|---|---|---|---|---|
| | $\boldsymbol{\delta} \sim U$ | $\boldsymbol{\delta} = \text{adv}$ | $\boldsymbol{\delta} \sim U$ | $\boldsymbol{\delta} = \text{adv}$ | $\boldsymbol{\delta} \sim U$ | $\boldsymbol{\delta} = \text{adv}$ |
| ARD | 0.2392 | 0.2848 | 0.1822 | 0.2176 | 0.4030 | 0.4734 |
| +IGDM | **0.1312** | **0.1533** | **0.1301** | **0.1501** | **0.1504** | **0.1690** |
| IAD | 0.2374 | 0.2811 | 0.1905 | 0.2190 | 0.3939 | 0.4568 |
| +IGDM | **0.1342** | **0.1684** | **0.1194** | **0.1395** | **0.1524** | **0.1971** |
| AKD | 0.3184 | 0.3622 | 0.2017 | 0.2323 | 0.4325 | 0.4810 |
| +IGDM | **0.1704** | **0.2081** | **0.1156** | **0.1312** | **0.1521** | **0.1698** |
| RSLAD | 0.1903 | 0.2375 | 0.1686 | 0.2019 | 0.2227 | 0.2856 |
| +IGDM | **0.1233** | **0.1516** | **0.1219** | **0.1424** | **0.1361** | **0.1681** |
| AdaAD | 0.1794 | 0.2529 | 0.1348 | 0.2028 | 0.2499 | 0.3400 |
| AdaIAD | 0.1803 | 0.2518 | 0.1331 | 0.2010 | 0.2504 | 0.3479 |
| +IGDM | **0.1294** | **0.1857** | **0.1253** | **0.1833** | **0.1338** | **0.2002** |

### 4.3 COMPARISON WITH THE STATE-OF-THE-ART METHOD: ADAAD

**Simple Inner Maximization with IGDM** AdaAD (Huang et al., 2023) employs a teacher model for inner maximization to enhance student model robustness. However, this approach significantly increases computational overhead due to the teacher model's large architecture. In contrast, IGDM achieves competitive robustness without a teacher model in inner maximization, simplifying the overall training process and reducing training time. In Table 6, we compare performance with and without the teacher model for inner maximization. When AdaAD's inner maximization is replaced with a PGD attack on the student model alone, robustness drops significantly. However, $\text{IGDM}_{\text{AdaAD}}$ achieves superior robustness more than the original AdaAD method, with reduced training time by

a factor of three. This highlights the efficiency of IGDM, offering both robustness and flexibility by eliminating the need for a teacher model on inner maximization.

Table 6: AutoAttack accuracy (%) and computational overhead of AdaAD and IGDM$_{\text{AdaAD}}$ with different inner loss on CIFAR-100 with the ResNet-18 student. 'T/E' and 'Mem' refer to time per epoch (in minutes) and memory usage. The 'w/o $T_{in}$' indicates that the inner loss is computed using a PGD attack solely on the student model, without the teacher's involvement.

| Method | BDM-AT teacher | | | LTD teacher | | | IKL-AT teacher | | |
|---|---|---|---|---|---|---|---|---|---|
| | AA | T/E⇓ | Mem⇓ | AA | T/E⇓ | Mem⇓ | AA | T/E⇓ | Mem⇓ |
| AdaAD | 28.06 | 10.62 | 4711M | 27.81 | 12.12 | 4900M | 26.89 | 10.91 | 4711M |
| AdaAD w/o $T_{in}$ | 25.62 | 2.26 | 2063M | 26.57 | 2.42 | 2634M | 23.37 | 2.27 | 2063M |
| IGDM$_{\text{AdaAD}}$ | 30.32 | 11.21 | 4711M | 28.87 | 13.27 | 4900M | 29.22 | 11.30 | 4711M |
| IGDM$_{\text{AdaAD}}$ w/o $T_{in}$ | 29.34 | 3.36 | 2611M | 28.12 | 3.64 | 3187M | 28.94 | 3.38 | 2611M |

**Comparison of IGDM and AdaAD under Same Experimental Conditions**  We replicated AdaAD's experimental setup using the same teacher-student model configuration for a fair comparison. Specifically, we used the CIFAR-100 dataset with the LTD teacher and the CIFAR-10 dataset with the LTD$_2$ teacher. (In the AdaAD paper, the LTD teacher was trained on WideResNet-34-20 for CIFAR-10, so we refer to it as LTD$_2$ to reflect this difference.) Both setups were evaluated with a ResNet-18 student, as shown in Table 7. While the AdaAD paper proposed two methods, AdaAD and AdaIAD, and noted a performance gap of about 1 percentage point, our experiments showed a smaller difference. As a result, we focused on applying the module specifically to AdaAD in our implementation. Notably, IGDM$_{\text{AdaAD}}$ consistently demonstrated a significant improvement in robustness accuracy. This demonstrates that even in a fair comparison, the IGDM module consistently enhances adversarial robustness, underscoring its effectiveness.

Table 7: Comparison between AdaAD paper results and our implementation results with identical experimental settings.

| Method | Result from | | CIFAR-100 with LTD | | CIFAR-10 with LTD$_2$ | |
|---|---|---|---|---|---|---|
| | Huang et al. (2023) | Ours | Clean | AA | Clean | AA |
| AdaAD | ✓ | | 62.19 | 26.74 | 85.58 | 51.37 |
| AdaIAD | ✓ | | 62.49 | 27.98 | 85.04 | 52.96 |
| AdaAD | | ✓ | 63.34 | 27.81 | 85.47 | 52.47 |
| AdaIAD | | ✓ | 63.24 | 27.83 | 85.34 | 52.30 |
| IGDM$_{\text{AdaAD}}$ | | ✓ | 63.44 | **28.87** | 85.50 | **53.45** |

**Limitations of Hyperparameter Tuning in AdaAD for Gradient Alignment**  AdaAD utilizes a single hyperparameter, $\lambda$, to control the balance between the distillation of adversarial and clean inputs, as $\lambda \cdot \text{KL}(f_S(\mathbf{x} + \boldsymbol{\delta}) \| f_T(\mathbf{x} + \boldsymbol{\delta})) + (1 - \lambda) \cdot \text{KL}(f_S(\mathbf{x}) \| f_T(\mathbf{x}))$. However, in their implementation, $\lambda$ is always set to one. Initially, we hypothesized that adjusting $\lambda$ could facilitate the distillation of both adversarial and clean inputs, potentially aligning the gradients. Yet, as shown in Table 8,

Table 8: Comparison between AdaAD with various hyperparameter.

| Method | Clean | AA | GD⇓ | GC⇑ |
|---|---|---|---|---|
| AdaAD($\lambda = 0.25$) | 66.48 | 25.51 | 0.118 | 0.526 |
| AdaAD($\lambda = 0.50$) | 65.44 | 27.24 | 0.108 | 0.564 |
| AdaAD($\lambda = 0.75$) | 64.98 | 27.42 | 0.109 | 0.560 |
| AdaAD($\lambda = 1.0$) | 64.43 | 27.82 | 0.107 | 0.568 |
| IGDM$_{\text{AdaAD}}$ | 64.44 | **30.32** | **0.086** | **0.643** |

modifying $\lambda$ degraded both robustness and gradient matching. This suggests that while point-wise matching is achieved in AdaAD, simply distilling two points is insufficient for capturing the teacher's gradients. In contrast, our approach enables effective gradient alignment and better distillation of the teacher model's robustness.

## 4.4 ABLATION STUDIES

In this section, we conduct more extensive experiments, including robustness against unseen attacks and distillation results with various teacher models. Here, we chose the CIFAR-100 dataset with ResNet-18 architecture of student model.

**Robustness against Unseen Attacks** In Table 9, we measured performance against the different attack scenarios in out-of-distribution (OOD) to demonstrate that we effectively distill the robustness of a teacher model. OODRobustBench (Li et al., 2023) is designed to simulate real-world distribution shifts and evaluate adversarial robustness. It focuses on two types of shifts: dataset shifts ($OOD_d$) and threat shifts ($OOD_t$), offering a more comprehensive assessment compared to relying solely on AutoAttack accuracy. The term $OOD_d$ encompasses natural and corruption shifts, which consist of variant datasets of CIFAR-100 and common corruptions such as noise and blur. Meanwhile, $OOD_t$ considers six unforeseen attacks, such as the Recolor (Laidlaw & Feizi, 2019) and StAdv (Xiao et al., 2018), etc. Overall, our module improves the performance of existing adversarial distillation methods against various noise and unseen attacks.

Table 9: The CIFAR-100 performance, evaluated with OODRobustBench.

| Method | Clean Acc (%) | Robust Acc (%) | |
| --- | --- | --- | --- |
| | $OOD_d$ | $OOD_d$ | $OOD_t$ |
| ARD | 49.92 | 15.70 | 14.91 |
| **+ IGDM** | **50.65** | **16.98** | **17.36** |
| IAD | 48.75 | 16.21 | 15.03 |
| **+ IGDM** | **51.43** | **18.20** | **17.35** |
| AKD | 49.74 | 15.95 | 14.96 |
| **+ IGDM** | 49.83 | **18.52** | **17.07** |
| RSLAD | 48.96 | 17.41 | 16.46 |
| **+ IGDM** | **50.45** | **19.66** | **18.53** |
| AdaAD | 51.53 | 17.77 | 17.32 |
| AdaIAD | 51.18 | 17.75 | 17.26 |
| **+ IGDM** | 51.52 | **19.66** | **18.65** |

**Adversarial Distillation with Various Teacher Model** We conducted adversarial distillation using several teacher models to further validate the effectiveness of our approach. We utilized multiple teacher models that were adversarially trained on the CIFAR-100 dataset. All these models are publicly available in RobustBench (Croce et al., 2021), and we selected them based on varying levels of robustness against AutoAttack (Croce & Hein, 2020a). As shown in the Figure 4, IGDM consistently outperforms other distillation methods across different teacher setups. Notably, even when using less robust teachers, there is a substantial performance gap. In contrast, while AdaAD demonstrates significant improvements with strong teachers compared to other baselines, it shows minimal differences when employing weaker teachers. This highlights that our approach maintains meaningful performance advantages across all teacher models.

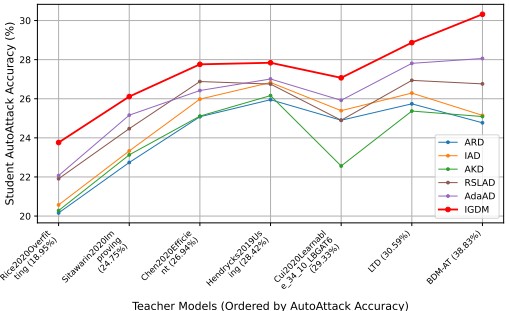

Figure 4: Performance comparison of different adversarial distillation methods across various teacher models.

## 5 CONCLUSION

We have proposed a novel method Indirect Gradient Distillation Module (IGDM) for adversarial distillation. In contrast to conventional adversarial distillation methods that primarily focus on distilling the logits of the teacher model, we concentrate on distilling the gradient information of the teacher model. We obtain these gradients indirectly by leveraging the locally linear property, a characteristic of adversarially trained models. Notably, IGDM can be seamlessly applied to existing adversarial distillation methods. Extensive experimental results demonstrate that the student model with IGDM successfully follows the gradients of the teacher model, resulting in significantly enhanced robustness.

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

# A FURTHER EXPERIMENTS

We conduct further experiments to corroborate our main contribution. These include distillation results on MobileNetV2 student (Appendix A.1), distillation with IKL-AT teacher (Appendix A.2), and experiments on the Tiny-ImagNet dataset (Appendix A.3). We also analyze the drawbacks of direct gradient matching (Appendix A.4) and the role of logit difference (Appendix A.5). Finally, hyperparameter tuning experiments are detailed in Appendix A.6.

## A.1 ADVERSARIAL DISTILLATION ON MOBILENETV2 ARCHITECTURE

Table 10: Adversarial distillation results on MobileNetV2 with BDM-AT and LTD teacher models on CIFAR-100. Bold indicates cases where IGDM improved accuracy by more than 0.5 percentage points or better gradient matching.

| Method | CIFAR-100 with BDM-AT teacher | | | | | CIFAR-100 with LTD teacher | | | | |
|---|---|---|---|---|---|---|---|---|---|---|
| | Clean | PGD | AA | GD⇓ | GC⇑ | Clean | PGD | AA | GD⇓ | GC⇑ |
| PGD-AT | 59.23 | 24.04 | 21.58 | 0.204 | 0.408 | 59.23 | 24.04 | 21.58 | 0.194 | 0.486 |
| +IGDM | 59.52 | **32.88** | **27.28** | **0.109** | **0.528** | 59.70 | **32.59** | **27.14** | **0.091** | **0.636** |
| TRADES | 51.05 | 24.83 | 20.62 | 0.132 | 0.377 | 51.05 | 24.83 | 20.62 | 0.119 | 0.460 |
| +IGDM | **57.80** | **29.47** | **22.05** | **0.097** | **0.469** | **56.81** | **29.87** | **23.38** | **0.081** | **0.640** |
| ARD | 60.74 | 29.92 | 24.33 | 0.137 | 0.445 | 60.55 | 30.82 | 25.28 | 0.111 | 0.569 |
| +IGDM | 60.83 | **33.55** | **27.50** | **0.107** | **0.534** | 60.44 | **33.36** | **27.59** | **0.082** | **0.681** |
| IAD | 56.35 | 28.96 | 23.43 | 0.129 | 0.430 | 56.11 | 29.55 | 24.22 | 0.106 | 0.547 |
| +IGDM | **57.65** | **31.97** | **25.60** | **0.106** | **0.506** | **58.60** | **31.36** | **25.64** | **0.090** | **0.617** |
| AKD | 60.84 | 29.37 | 24.22 | 0.135 | 0.428 | 60.65 | 29.84 | 24.85 | 0.112 | 0.564 |
| +IGDM | 60.39 | **33.62** | **28.04** | **0.104** | **0.538** | 59.94 | **33.70** | **28.12** | **0.077** | **0.715** |
| RSLAD | 61.29 | 31.74 | 26.18 | 0.121 | 0.490 | 60.43 | 32.37 | 26.85 | 0.092 | 0.632 |
| +IGDM | 61.48 | **35.22** | **29.32** | **0.101** | **0.560** | 60.36 | **33.99** | **28.00** | **0.079** | **0.696** |
| AdaAD | 61.89 | 29.54 | 23.80 | 0.118 | 0.497 | 61.43 | 30.58 | 25.03 | 0.091 | 0.649 |
| AdaIAD | 60.83 | 29.94 | 23.83 | 0.116 | 0.499 | 61.22 | 30.78 | 25.22 | 0.088 | 0.652 |
| +IGDM | 61.43 | **33.51** | **27.43** | **0.097** | **0.583** | 61.57 | **33.14** | **27.65** | **0.076** | **0.721** |

Table 11: Adversarial distillation results on MobileNetV2 with BDM-AT and LTD teacher models on CIFAR-10. Bold indicates cases where IGDM improved accuracy by more than 0.5 percentage points or better gradient matching.

| Method | CIFAR-10 with BDM-AT teacher | | | | | CIFAR-10 with LTD teacher | | | | |
|---|---|---|---|---|---|---|---|---|---|---|
| | Clean | PGD | AA | GD⇓ | GC⇑ | Clean | PGD | AA | GD⇓ | GC⇑ |
| PGD-AT | 83.52 | 44.47 | 41.19 | 0.175 | 0.406 | 83.52 | 44.47 | 41.19 | 0.171 | 0.505 |
| +IGDM | 82.78 | **51.54** | **47.13** | **0.074** | **0.501** | 81.73 | **52.63** | **48.51** | **0.048** | **0.671** |
| TRADES | 81.57 | 50.49 | 46.88 | 0.074 | 0.492 | 81.57 | 50.49 | 46.88 | 0.062 | 0.605 |
| +IGDM | **82.33** | **52.94** | **47.70** | **0.050** | **0.502** | 81.82 | **52.96** | **47.47** | **0.039** | **0.645** |
| ARD | 84.38 | 48.26 | 44.02 | 0.091 | 0.371 | 84.18 | 52.16 | 48.11 | 0.054 | 0.578 |
| +IGDM | 84.53 | **52.45** | **45.06** | **0.071** | **0.467** | 84.25 | **54.03** | **49.53** | **0.042** | **0.684** |
| IAD | 83.79 | 48.36 | 44.02 | 0.088 | 0.385 | 83.38 | 52.71 | 48.45 | 0.051 | 0.592 |
| +IGDM | 84.27 | **52.42** | **45.08** | **0.063** | **0.497** | 83.65 | **54.57** | **50.14** | **0.038** | **0.704** |
| AKD | 83.75 | 47.15 | 43.43 | 0.093 | 0.368 | 84.46 | 50.84 | 46.96 | 0.057 | 0.566 |
| +IGDM | 83.29 | **53.29** | **48.41** | **0.064** | **0.527** | 83.98 | **54.40** | **49.99** | **0.040** | **0.691** |
| RSLAD | 84.69 | 50.94 | 47.38 | 0.074 | 0.491 | 82.60 | 54.55 | 50.44 | 0.040 | 0.706 |
| +IGDM | 84.64 | **52.33** | **49.27** | **0.068** | **0.515** | 82.61 | **55.02** | **51.01** | **0.039** | **0.706** |
| AdaAD | 84.51 | 50.67 | 46.56 | 0.069 | 0.516 | 84.24 | 55.47 | 51.36 | 0.034 | 0.777 |
| AdaIAD | 85.04 | 51.33 | 47.62 | 0.068 | 0.519 | 84.14 | 55.58 | 51.48 | 0.034 | 0.778 |
| +IGDM | 84.75 | **52.35** | **48.34** | **0.065** | **0.526** | 84.27 | **56.15** | **52.15** | **0.033** | **0.790** |

In Table 10 and Table 11, we present additional experiments on the CIFAR-10 and CIFAR-100 datasets using MobileNetV2 (Sandler et al., 2018) as the student model, with two distinct teacher models: LTD and BDM-AT. Across both datasets, applying IGDM consistently improved robustness under PGD and AutoAttack, compared to the baseline adversarial training methods. Notably, IGDM enhanced the model's performance on both CIFAR-10 and CIFAR-100, particularly in reducing the Gradient Distance (GD) and improving Gradient Cosine similarity (GC). These results underscore the effectiveness of IGDM in improving adversarial robustness and gradient alignment across different datasets and architectures.

## A.2 ADVERSARIAL DISTILLATION WITH IKL-AT TEACHER MODEL.

Table 12: Adversarial distillation results on ResNet-18 and MobileNetV2 with a IKL-AT teacher model on CIFAR-100. Bold indicates cases where IGDM improved accuracy by more than 0.5 percentage points or better gradient matching.

| Method | CIFAR-100 with ResNet-18 student | | | | | CIFAR-100 with MobileNetV2 student | | | | |
|---|---|---|---|---|---|---|---|---|---|---|
| | Clean | PGD | AA | GD⇓ | GC⇑ | Clean | PGD | AA | GD⇓ | GC⇑ |
| PGD-AT | 55.80 | 19.88 | 18.86 | 0.460 | 0.306 | 59.23 | 24.04 | 21.58 | 0.211 | 0.392 |
| **+IGDM** | **62.91** | **35.25** | **28.89** | **0.122** | **0.534** | **59.75** | **33.59** | **27.02** | **0.120** | **0.517** |
| TRADES | 53.56 | 25.85 | 22.02 | 0.199 | 0.354 | 51.05 | 24.83 | 20.62 | 0.148 | 0.351 |
| **+IGDM** | **62.41** | **31.79** | **24.02** | **0.116** | **0.467** | **59.34** | **29.36** | **22.40** | **0.120** | **0.444** |
| ARD | 61.38 | 27.59 | 23.18 | 0.178 | 0.400 | 61.52 | 28.00 | 23.40 | 0.164 | 0.420 |
| **+IGDM** | 61.55 | **35.24** | **28.87** | **0.117** | **0.546** | 61.95 | **32.26** | **26.48** | **0.132** | **0.498** |
| IAD | 61.09 | 29.25 | 23.61 | 0.173 | 0.403 | 58.39 | 28.40 | 23.06 | 0.147 | 0.418 |
| **+IGDM** | **64.20** | **33.71** | **27.03** | **0.132** | **0.506** | **59.58** | **30.96** | **25.42** | **0.128** | **0.478** |
| AKD | 61.90 | 29.23 | 23.96 | 0.171 | 0.412 | 61.69 | 27.87 | 23.60 | 0.159 | 0.431 |
| **+IGDM** | 61.94 | **33.16** | **27.82** | **0.131** | **0.511** | 61.56 | **33.62** | **27.32** | **0.117** | **0.519** |
| RSLAD | 61.18 | 30.54 | 25.27 | 0.147 | 0.444 | 61.95 | 30.16 | 25.09 | 0.157 | 0.441 |
| **+IGDM** | **63.55** | **35.26** | **29.10** | **0.119** | **0.541** | **62.48** | **35.25** | **28.82** | **0.118** | **0.547** |
| AdaAD | 65.36 | 32.29 | 26.89 | 0.133 | 0.525 | 62.35 | 28.51 | 23.01 | 0.140 | 0.470 |
| AdaIAD | 65.29 | 32.47 | 26.80 | 0.133 | 0.527 | 62.17 | 28.48 | 23.39 | 0.139 | 0.473 |
| **+IGDM** | **66.00** | **34.47** | **29.22** | **0.119** | **0.582** | 62.02 | **32.85** | **26.11** | **0.119** | **0.533** |

Table 13: Adversarial distillation results on ResNet-18 and MobileNetV2 with a IKL-AT teacher model on CIFAR-10. Bold indicates cases where IGDM improved accuracy by more than 0.5 percentage points or better gradient matching.

| Method | CIFAR-10 with ResNet-18 student | | | | | CIFAR-10 with MobileNetV2 student | | | | |
|---|---|---|---|---|---|---|---|---|---|---|
| | Clean | PGD | AA | GD⇓ | GC⇑ | Clean | PGD | AA | GD⇓ | GC⇑ |
| PGD-AT | 84.52 | 42.80 | 41.12 | 0.409 | 0.286 | 83.52 | 44.47 | 41.19 | 0.175 | 0.401 |
| **+IGDM** | 83.66 | **54.38** | **49.85** | **0.072** | **0.541** | 82.11 | **51.13** | **47.00** | **0.063** | **0.507** |
| TRADES | 82.46 | 49.13 | 47.09 | 0.131 | 0.425 | 81.57 | 50.49 | 46.88 | 0.073 | 0.489 |
| **+IGDM** | **83.14** | **54.91** | **48.76** | **0.098** | **0.455** | 81.67 | **52.32** | **47.40** | **0.057** | **0.492** |
| ARD | 85.41 | 49.36 | 45.32 | 0.087 | 0.389 | 84.68 | 48.03 | 44.11 | 0.087 | 0.376 |
| **+IGDM** | 85.78 | **54.37** | **46.52** | **0.072** | **0.467** | 84.72 | **50.89** | **46.13** | **0.070** | **0.473** |
| IAD | 85.22 | 49.70 | 45.96 | 0.083 | 0.393 | 84.25 | 48.58 | 44.15 | 0.082 | 0.390 |
| **+IGDM** | 83.78 | **54.77** | **50.24** | **0.061** | **0.521** | 83.89 | **51.46** | **46.48** | **0.069** | **0.458** |
| AKD | 83.62 | 51.81 | 46.01 | 0.080 | 0.395 | 84.52 | 47.56 | 43.27 | 0.154 | 0.421 |
| **+IGDM** | 83.70 | **53.69** | **49.45** | **0.067** | **0.539** | 84.33 | **50.36** | **45.37** | **0.089** | **0.438** |
| RSLAD | 85.55 | 52.13 | 48.83 | 0.069 | 0.531 | 84.62 | 51.43 | 47.69 | 0.072 | 0.503 |
| **+IGDM** | 85.67 | **53.30** | **49.58** | **0.065** | **0.541** | 84.30 | **53.14** | **48.91** | **0.066** | **0.523** |
| AdaAD | 86.59 | 54.25 | 50.86 | 0.062 | 0.570 | 85.31 | 52.10 | 48.03 | 0.066 | 0.524 |
| AdaIAD | 86.57 | 54.59 | 50.61 | 0.062 | 0.572 | 85.47 | 52.05 | 47.96 | 0.066 | 0.523 |
| **+IGDM** | 86.10 | **55.41** | **51.29** | **0.060** | **0.587** | 85.11 | **53.07** | **49.08** | **0.061** | **0.541** |

In Table 12 and Table 13, we conduct additional experiments on the CIFAR-100 and CIFAR-10 datasets with the IKL-AT (Cui et al., 2023) teacher model. Similar to the outcomes observed with the LTD (Chen & Lee, 2021) and BDM-AT (Wang et al., 2023b) teachers on those datasets, IGDM demonstrates substantial enhancements in robustness with the IKL-AT teacher, regardless of the original methods, datasets, or student models, while maintaining consistent clean accuracy. Furthermore, consistent results are observed across all experiments, with a decrease in GD and an increase in GC upon applying IGDM.

## A.3 ADVERSARIAL DISTILLATION ON TINY-IMAGENET

Table 14: Adversarial distillation result on PreActResNet-18 with a WideResNet-34-10 teacher model on Tiny-ImageNet. Bold indicates cases where IGDM improved accuracy by more than 0.5 percentage points or better gradient matching.

| Method | Clean | PGD | AA | GD$\Downarrow$ | GC$\Uparrow$ |
|---|---|---|---|---|---|
| PGD-AT (Madry et al., 2017) | 50.13 | 17.10 | 14.58 | 0.126 | 0.490 |
| **+IGDM** | **51.46** | **25.04** | **20.10** | **0.071** | **0.672** |
| TRADES (Zhang et al., 2019) | 46.51 | 19.51 | 14.89 | 0.094 | 0.447 |
| **+IGDM** | **51.31** | **24.10** | **18.92** | **0.073** | **0.649** |
| ARD (Goldblum et al., 2020) | 50.81 | 23.37 | 19.48 | 0.081 | 0.613 |
| **+IGDM** | 50.86 | **25.39** | **20.57** | **0.069** | **0.683** |
| IAD (Zhu et al., 2021) | 49.89 | 23.89 | 19.10 | 0.079 | 0.614 |
| **+IGDM** | 50.02 | **25.57** | **20.69** | **0.066** | **0.704** |
| AKD (Maroto et al., 2022) | 51.70 | 23.86 | 19.69 | 0.082 | 0.612 |
| **+IGDM** | 51.38 | **25.25** | **20.99** | **0.063** | **0.720** |
| RSLAD (Zi et al., 2021) | 47.12 | 22.14 | 17.65 | 0.081 | 0.567 |
| **+IGDM** | 47.54 | **23.95** | **18.35** | **0.074** | **0.620** |
| AdaAD (Huang et al., 2023) | 51.54 | 24.65 | 20.60 | 0.064 | 0.738 |
| **+IGDM** | 51.21 | **25.73** | **21.17** | **0.060** | **0.764** |

The results in Table 14 confirm the effectiveness of IGDM in enhancing adversarial robustness, especially on the challenging Tiny-ImageNet dataset Le & Yang (2015). Consistent with findings on other datasets, IGDM improves robustness metrics such as PGD and AA accuracy, while maintaining comparable clean accuracy. This improvement is particularly significant given the complexity of Tiny-ImageNet, where achieving high robustness is often challenging. Moreover, IGDM facilitates better alignment between teacher and student models, evidenced by a reduction in mean Gradient Distance (GD) and an increase in mean Gradient Cosine similarity (GC), further underscoring the impact of gradient matching of our method.

## A.4 DRAWBACKS OF DIRECT GRADIENT MATCHING IN CONTRAST TO IGDM

IGDM achieves gradient matching indirectly by leveraging the differences in logits. One might assume that distilling directly calculated gradients could achieve gradient matching more intuitively, as expressed by the following equation:

$$L_{Direct} = T(\alpha) \cdot D \left( \frac{\partial f_S(\mathbf{x})}{\partial \mathbf{x}} , \frac{\partial f_T(\mathbf{x})}{\partial \mathbf{x}} \right). \tag{11}$$

In Table 15, we conduct experiment on IGDM$_{ARD}$ and direct gradient matching (Direct$_{ARD}$). To ensure a fair comparison between the two methods, we kept all other factors the same, varying only the gradient calculation method (logits differences vs. direct calculation). The results reveal that direct gradient distillation fails to meaningfully enhance robustness compared to the original ARD method. Although it shows a slight enhancement in gradient matching, its performance significantly lags behind IGDM.

To provide a comprehensive interpretation, we analyze GD and training loss over training time, epoch, and step in Figure 5. The limitations of direct gradient matching are evident due to optimization challenges; a small $\alpha$ fails to match the gradient, while a large $\alpha$ results in poor convergence of

Table 15: Adversarial distillation result of ARD variant methods on ResNet-18 with BDM-AT teacher on CIFAR-100. Direct$_{ARD}$ represents the direct distillation of gradients, and the $\alpha$ is the hyperparameter of the gradient distillation loss.

| Method | Clean | PGD | AA | GD$\Downarrow$ | GC$\Uparrow$ |
|---|---|---|---|---|---|
| Direct$_{ARD}(\alpha = 1)$ | 60.92 | 30.56 | 25.23 | 0.136 | 0.447 |
| Direct$_{ARD}(\alpha = 3)$ | 60.98 | 30.29 | 24.88 | 0.137 | 0.444 |
| Direct$_{ARD}(\alpha = 6)$ | 61.13 | 30.34 | 25.27 | 0.135 | 0.448 |
| Direct$_{ARD}(\alpha = 10)$ | 61.05 | 30.80 | 25.33 | 0.132 | 0.453 |
| Direct$_{ARD}(\alpha = 30)$ | 61.43 | 30.45 | 25.29 | 0.127 | 0.455 |
| Direct$_{ARD}(\alpha = 60)$ | 61.14 | 30.17 | 24.73 | 0.124 | 0.450 |
| Direct$_{ARD}(\alpha = 100)$ | 60.36 | 30.48 | 24.62 | 0.121 | 0.445 |
| Direct$_{ARD}(\alpha = 300)$ | 59.63 | 29.72 | 23.99 | 0.116 | 0.440 |
| Direct$_{ARD}(\alpha = 600)$ | 58.79 | 29.76 | 24.06 | 0.111 | 0.441 |
| Direct$_{ARD}(\alpha = 1000)$ | 57.08 | 29.89 | 24.14 | 0.108 | 0.445 |
| Direct$_{ARD}(\alpha = 3000)$ | 51.30 | 29.13 | 22.85 | 0.104 | 0.441 |
| Direct$_{ARD}(\alpha = 6000)$ | 46.95 | 27.33 | 20.70 | 0.105 | 0.424 |
| Direct$_{ARD}(\alpha = 10000)$ | 43.22 | 25.67 | 19.18 | 0.105 | 0.404 |
| ARD (Goldblum et al., 2020) | 61.51 | 30.23 | 24.77 | 0.142 | 0.439 |
| **+IGDM** | **61.62** | **35.75** | **28.79** | **0.101** | **0.571** |

the training loss. More interestingly, even with a large $\alpha$, direct matching results in GD values higher than those achieved by IGDM. We interpret these optimization issues as arising from the low-level features of the input gradient, similar to the input itself. In other words, directly matching low-level feature gradients is comparable to matching pixel-wise image differences. This leads to the model's inability to effectively capture the training objective, failing to match the gradient and, consequently, diminished robustness. One potential solution to address these optimization challenges is to employ an additional discriminator model, as proposed in (Chan et al., 2020). However, this approach requires training an additional discriminator model, whereas IGDM does not necessitate any other model or training procedure. In summary, IGDM significantly enhances robustness by indirectly matching the gradient using high-level feature logits, thereby achieving superior robustness.

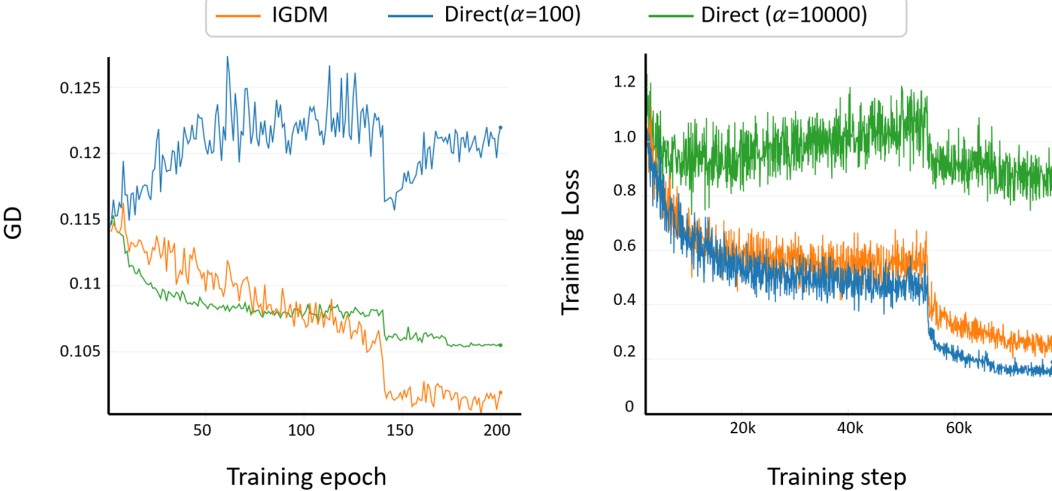

Figure 5: Comparision between IGDM and direct gradient matching. GD and training loss are measured at the training epoch and step, respectively.

A.5 IMPORTANT ROLE OF LOGIT DIFFERENCE IN IGDM

Adversarial training aims to make the network output almost constant within a ball around the input point. Therefore, one might assume that the distilling logit difference of the adversarially trained teacher model in IGDM loss is close to distilling 0. In other words, IGDM can be simplified as expressed by the following equation:

$$
\begin{aligned}
L_{\text{IGDM}_{\text{TRADES-like}}} &= T(\alpha) \cdot D(f_S(\mathbf{x} + \boldsymbol{\delta}) - f_S(\mathbf{x} - \boldsymbol{\delta}) \,,\, \mathbf{0}) \\
&= T(\alpha) \cdot D(f_S(\mathbf{x} + \boldsymbol{\delta}) \,,\, f_S(\mathbf{x} - \boldsymbol{\delta})).
\end{aligned}
\tag{12}
$$

We refer to this method as IGDM$_{\text{TRADES-like}}$ because it resembles a TRADES (Zhang et al., 2019) regularization loss (TRADES$_{\text{reg}}$), and empirically, its performance closely aligns with the addition of the TRADES regularization loss:

$$
L_{\text{TRADES}_{\text{reg}}} = T(\alpha) \cdot D(f_S(\mathbf{x} + \boldsymbol{\delta}) \; f_S(\mathbf{x})).
\tag{13}
$$

Table 16: Adversarial distillation result of ARD variant methods on ResNet-18 with BDM-AT teacher on CIFAR-100. TRADES$_{\text{reg}}$ represents the regularization loss of TRADES. and the $\alpha$ is the hyperparameter of the gradient distillation loss.

| Method | Clean | PGD | AA | GD⇓ | GC⇑ |
|---|---|---|---|---|---|
| ARD + TRADES$_{\text{reg}}(\alpha = 1)$ | 57.86 | 31.44 | 25.95 | 0.125 | 0.459 |
| ARD + TRADES$_{\text{reg}}(\alpha = 5)$ | 58.07 | 32.13 | 26.06 | 0.117 | 0.466 |
| ARD + TRADES$_{\text{reg}}(\alpha = 10)$ | 53.58 | 32.01 | 26.27 | 0.116 | 0.463 |
| ARD + TRADES$_{\text{reg}}(\alpha = 15)$ | 52.05 | 31.53 | 26.23 | 0.115 | 0.464 |
| ARD + TRADES$_{\text{reg}}(\alpha = 20)$ | 50.87 | 31.92 | 26.30 | 0.115 | 0.462 |
| ARD + TRADES$_{\text{reg}}(\alpha = 25)$ | 50.28 | 31.92 | 26.15 | 0.115 | 0.458 |
| ARD + TRADES$_{\text{reg}}(\alpha = 30)$ | 49.20 | 31.93 | 25.87 | 0.115 | 0.456 |
| ARD + IGDM$_{\text{TRADES-like}}(\alpha = 1)$ | 57.73 | 31.72 | 25.64 | 0.119 | 0.461 |
| ARD + IGDM$_{\text{TRADES-like}}(\alpha = 5)$ | 54.18 | 31.71 | 25.79 | 0.117 | 0.466 |
| ARD + IGDM$_{\text{TRADES-like}}(\alpha = 10)$ | 52.21 | 32.08 | 25.98 | 0.118 | 0.464 |
| ARD + IGDM$_{\text{TRADES-like}}(\alpha = 15)$ | 50.50 | 31.85 | 25.91 | 0.118 | 0.457 |
| ARD + IGDM$_{\text{TRADES-like}}(\alpha = 20)$ | 49.67 | 31.76 | 25.78 | 0.118 | 0.452 |
| ARD + IGDM$_{\text{TRADES-like}}(\alpha = 25)$ | 48.21 | 31.33 | 25.38 | 0.119 | 0.445 |
| ARD + IGDM$_{\text{TRADES-like}}(\alpha = 30)$ | 47.61 | 31.04 | 25.05 | 0.119 | 0.439 |
| ARD (Goldblum et al., 2020) | 61.51 | 30.23 | 24.77 | 0.142 | 0.439 |
| **+IGDM** | **61.62** | **35.75** | **28.79** | **0.101** | **0.571** |

In Table 16, we conduct experiments on these two methods in conjunction with the ARD (Goldblum et al., 2020) method. For TRADES$_{\text{reg}}$, as the regularization term strengthens, the robustness slightly increases; however, it lags far behind IGDM in terms of robustness. IGDM$_{\text{TRADES-like}}$ follows a similar trend to the TRADES$_{\text{reg}}$ method with the increase in hyperparameter. These results indicate that the logit difference of the teacher model provides valuable information for teaching the student model to achieve robustness through gradient matching, resulting in superior robustness. This finding corroborates our main contribution in the main paper.

A.6 ANALYSIS ON HYPERPARAMETER SENSITIVITY OF IGDM

The proposed IGDM loss is defined as follows:

$$
L_{IGDM} = T(\alpha) \cdot D\left(f_S(\mathbf{x} + \boldsymbol{\delta}) - f_S(\mathbf{x} - \boldsymbol{\delta}) \,,\, f_T(\mathbf{x} + \boldsymbol{\delta}) - f_T(\mathbf{x} - \boldsymbol{\delta})\right).
$$

In Table 17 and Table 18, we analyze the sensitivity of hyperparameters $\alpha$ using IGDM$_{\text{PGD-AT}}$ and IGDM$_{\text{ARD}}$. In these tables, a value of 0 for $\alpha$ represents the original methods, PGD-AT and ARD, respectively. As $\alpha$ increases, gradient matching becomes more prominent, leading to enhanced robustness for both methods. However, excessively large values of $\alpha$ do not provide additional enhancements and may even reduce robustness. Based on the insights from the hyperparameter sensitivity analysis, we employed grid search to adapt the IGDM module for all experiments.

Table 17: Analysis of hyperparameter of IGDM using PGD-AT variant methods on ResNet-18 with BDM-AT teacher on CIFAR-100. $\alpha$ is the hyperparameter of IGDM loss. The gray row indicates the reported values.

| Method | Clean | PGD | AA | GD$\Downarrow$ | GC$\Uparrow$ |
|---|---|---|---|---|---|
| PGD-AT (Madry et al., 2017) | 55.80 | 19.88 | 18.86 | 0.452 | 0.389 |
| IGDM$_{\text{PGD-AT}}(\alpha = 20)$ | 56.95 | 28.10 | 24.02 | 0.157 | 0.427 |
| IGDM$_{\text{PGD-AT}}(\alpha = 40)$ | 58.23 | 30.36 | 25.71 | 0.139 | 0.457 |
| IGDM$_{\text{PGD-AT}}(\alpha = 60)$ | 58.74 | 31.69 | 26.57 | 0.131 | 0.483 |
| IGDM$_{\text{PGD-AT}}(\alpha = 80)$ | 59.01 | 32.02 | 27.08 | 0.123 | 0.504 |
| IGDM$_{\text{PGD-AT}}(\alpha = 100)$ | 59.20 | 33.09 | 27.48 | 0.118 | 0.519 |
| IGDM$_{\text{PGD-AT}}(\alpha = 120)$ | 59.42 | 33.60 | 27.95 | 0.114 | 0.531 |
| IGDM$_{\text{PGD-AT}}(\alpha = 140)$ | 59.32 | 34.18 | 28.26 | 0.111 | 0.542 |
| IGDM$_{\text{PGD-AT}}(\alpha = 160)$ | 60.06 | 34.38 | 28.47 | 0.108 | 0.550 |
| IGDM$_{\text{PGD-AT}}(\alpha = 180)$ | 60.53 | 34.47 | 28.51 | 0.105 | 0.556 |
| IGDM$_{\text{PGD-AT}}(\alpha = 200)$ | 60.83 | 34.90 | 28.84 | 0.104 | 0.562 |
| IGDM$_{\text{PGD-AT}}(\alpha = 220)$ | 60.68 | 35.09 | **29.16** | 0.102 | 0.569 |
| IGDM$_{\text{PGD-AT}}(\alpha = 240)$ | 59.65 | 35.34 | 29.10 | 0.101 | 0.572 |
| IGDM$_{\text{PGD-AT}}(\alpha = 260)$ | 59.71 | 35.37 | 29.09 | 0.100 | 0.573 |
| IGDM$_{\text{PGD-AT}}(\alpha = 280)$ | 59.83 | 35.23 | 28.84 | 0.099 | 0.576 |
| IGDM$_{\text{PGD-AT}}(\alpha = 300)$ | 59.95 | 35.39 | 29.13 | 0.098 | 0.579 |

Table 18: Analysis of hyperparameter of IGDM using ARD variant methods on ResNet-18 with BDM-AT teacher on CIFAR-100. $\alpha$ is the hyperparameter of IGDM loss. The gray row indicates the reported values.

| Method | Clean | PGD | AA | GD$\Downarrow$ | GC$\Uparrow$ |
|---|---|---|---|---|---|
| ARD (Goldblum et al., 2020) | 61.51 | 30.23 | 24.77 | 0.1422 | 0.439 |
| IGDM$_{\text{ARD}}(\alpha = 20)$ | 59.74 | 33.97 | 27.85 | 0.112 | 0.523 |
| IGDM$_{\text{ARD}}(\alpha = 40)$ | 60.13 | 34.21 | 28.56 | 0.106 | 0.546 |
| IGDM$_{\text{ARD}}(\alpha = 60)$ | 61.08 | 34.54 | 28.42 | 0.104 | 0.557 |
| IGDM$_{\text{ARD}}(\alpha = 80)$ | 60.78 | 35.21 | 28.73 | 0.102 | 0.568 |
| IGDM$_{\text{ARD}}(\alpha = 100)$ | 61.18 | 35.75 | **28.79** | 0.101 | 0.571 |
| IGDM$_{\text{ARD}}(\alpha = 120)$ | 61.15 | 35.17 | 28.67 | 0.099 | 0.578 |
| IGDM$_{\text{ARD}}(\alpha = 140)$ | 61.24 | 35.32 | 28.75 | 0.098 | 0.583 |
| IGDM$_{\text{ARD}}(\alpha = 160)$ | 60.76 | 35.81 | 28.71 | 0.096 | 0.584 |
| IGDM$_{\text{ARD}}(\alpha = 180)$ | 60.19 | 35.67 | 28.78 | 0.095 | 0.585 |
| IGDM$_{\text{ARD}}(\alpha = 200)$ | 59.95 | 35.19 | 28.67 | 0.095 | 0.585 |
| IGDM$_{\text{ARD}}(\alpha = 220)$ | 59.42 | 35.24 | 28.76 | 0.096 | 0.588 |
| IGDM$_{\text{ARD}}(\alpha = 240)$ | 59.76 | 34.78 | 28.74 | 0.095 | 0.587 |
| IGDM$_{\text{ARD}}(\alpha = 260)$ | 59.04 | 35.01 | 28.36 | 0.095 | 0.586 |
| IGDM$_{\text{ARD}}(\alpha = 280)$ | 59.56 | 35.21 | 28.59 | 0.095 | 0.589 |
| IGDM$_{\text{ARD}}(\alpha = 300)$ | 58.91 | 34.78 | 28.58 | 0.095 | 0.582 |

# B   TRAINING DETAILS

## B.1   SETTINGS

We utilized the CIFAR-10, CIFAR-100 (Krizhevsky et al., 2009), SVHN Netzer et al. (2011), and Tiny-ImageNet (Le & Yang, 2015) datasets with random crop and random horizontal flips, excluding other augmentations. We trained all AT, AD, and IGDM incorporated methods using an SGD momentum optimizer with the same initial learning rate of 0.1, momentum of 0.9, and weight decay of 5e-4.

For CIFAR-10 and CIFAR-100, we adhered to the training settings of other adversarial distillation methods, training for 200 epochs, except for RSLAD and IGDM$_{\text{RSLAD}}$, which were trained for 300 epochs. RSLAD suggested that increasing the number of training epochs could enhance model

robustness; thus, we followed their recommendation to train for 300 epochs. The learning rate scheduler reduced the learning rate by a factor of 10 at the 100th and 150th epochs. However, for RSLAD and IGDM$_{\text{RSLAD}}$, the learning rate decreased by 10 at the 215th, 260th, and 285th epochs, as suggested in the original paper. For SVHN, we trained for 50 epochs with the learning rate decayed by a factor of 10 at the 40th and 45th epochs for all methods. For Tiny-ImageNet, we trained for 100 epochs with the learning rate decayed by a factor of 10 at the 50th and 80th epochs for all methods.

The adversarial perturbation settings were as follows: the number of iterations for inner maximization was set to 10, with a step size of $2/255$, and a total perturbation bound of $L_\infty = 8/255$. Specifically, we employed the recommended inner loss functions as outlined in the original paper: PGD attack on student model for PGD-AT, ARD, and IAD; TRADES attack on student model for TRADES; RSLAD inner loss for RSLAD; and AdaAD inner loss for AdaAD. Moreover, the IGDM-incorporated method followed the inner maximization method of the original AT or AD method. For formulaic representation as follows,

$$\text{PGD attack: } \boldsymbol{\delta} = \underset{\|\boldsymbol{\delta'}\|_p \leq \epsilon}{\arg\max} \, \text{CE}(f_S(\mathbf{x} + \boldsymbol{\delta'}), y), \tag{14}$$

$$\text{TRADES attack: } \boldsymbol{\delta} = \underset{\|\boldsymbol{\delta'}\|_p \leq \epsilon}{\arg\max} \, \text{KL}(f_S(\mathbf{x} + \boldsymbol{\delta'}), f_S(\mathbf{x})), \tag{15}$$

$$\text{RSLAD attack: } \boldsymbol{\delta} = \underset{\|\boldsymbol{\delta'}\|_p \leq \epsilon}{\arg\max} \, \text{KL}(f_S(\mathbf{x} + \boldsymbol{\delta'}), f_T(\mathbf{x})), \tag{16}$$

$$\text{AdaAD attack: } \boldsymbol{\delta} = \underset{\|\boldsymbol{\delta'}\|_p \leq \epsilon}{\arg\max} \, \text{KL}(f_S(\mathbf{x} + \boldsymbol{\delta'}), f_T(\mathbf{x} + \boldsymbol{\delta'})). \tag{17}$$

Here, CE represents cross-entropy loss, KL represents KL-divergence loss.

## B.2 HYPERPARAMETER

The parameters of AT, AD, and the AD component in IGDM-incorporated methods were strictly set to the value suggested in the original paper. On the other hand, a parameter of IGDM varied depending on the original AT or AD method, dataset, teacher model, and student model. We experimentally employed an increasing hyperparameter function, defined as $T(\alpha) = \frac{\text{Current Epoch}}{\text{Total Epochs}} \cdot \alpha$, and the $\alpha$ value varied in each training scenario. The following paragraphs provide detailed information on the $\alpha$ in each experimental setting, determined through grid search results. IGDM$_{\text{AdaAD}}$ and IGDM$_{\text{AdaIAD}}$ have same value of hyperparameter in all cases.

### B.2.1 RESNET-18 MODEL TRAINED ON CIFAR-100 DATASET

For IGDM$_{\text{PGD-AT}}$, to 100 for LTD and IKL-AT, 220 for BDM-AT. For IGDM$_{\text{TRADES}}$, 5 for all teachers. For IGDM$_{\text{ARD}}$, 50 for LTD and 100 for BDM-AT and IKL-AT. For IGDM$_{\text{IAD}}$, 20 for LTD and 50 for BDM-AT and IKL-AT. For IGDM$_{\text{AKD}}$, 50 for LTD, 25 for BDM-AT, and 40 for IKL-AT. For IGDM$_{\text{RSLAD}}$, 3 for LTD and 10 for BDM-AT and IKL-AT. For IGDM$_{\text{AdaAD}}$, 15 for LTD, 70 for BDM-AT, and 50 for IKL-AT.

### B.2.2 RESNET-18 MODEL TRAINED ON THE CIFAR-10 DATASET

For IGDM$_{\text{PGD-AT}}$, 60 for LTD and 70 for BDM-AT and IKL-AT. For IGDM$_{\text{TRADES}}$, 20 for all teachers. For IGDM$_{\text{ARD}}$, 5 for LTD and 10 for BDM-AT and IKL-AT. For IGDM$_{\text{IAD}}$, 50 for LTD and 30 for BDM-AT and IKL-AT. For IGDM$_{\text{AKD}}$, 7 for LTD, 10 for BDM-AT, and 15 for IKL-AT. For IGDM$_{\text{RSLAD}}$, 0.9 for LTD and BDM-AT, and 1 for IKL-AT. For IGDM$_{\text{AdaAD}}$, 5 for LTD and IKL-AT, and 10 for BDM-AT.

### B.2.3 MOBILENETV2 MODEL TRAINED ON CIFAR-100 DATASET

For IGDM$_{\text{PGD-AT}}$, 150 for LTD, 160 for BDM-AT, and IKL-AT. For IGDM$_{\text{TRADES}}$, 10 for LTD and BDM-AT, and 3 for IKL-AT. For IGDM$_{\text{ARD}}$, 70 for all teachers. For IGDM$_{\text{IAD}}$, 20 for LTD and 50 for BDM-AT and IKL-AT. For IGDM$_{\text{AKD}}$, 30 for all teachers. For IGDM$_{\text{RSLAD}}$, 3 for LTD, 4 for BDM-AT, and 10 for IKL-AT. For IGDM$_{\text{AdaAD}}$, 20 for LTD, 50 for BDM-AT, and 40 for IKL-AT.

### B.2.4  MobileNetV2 model trained on CIFAR-10 dataset

For $\text{IGDM}_{\text{PGD-AT}}$, 70 for LTD, 40 for BDM-AT, and 50 for IKL-AT. For $\text{IGDM}_{\text{TRADES}}$, 1 for LTD, 0.5 for BDM-AT, and 0.2 for IKL-AT. For $\text{IGDM}_{\text{ARD}}$, 10 for LTD, 8 for BDM-AT, and 9 for IKL-AT. For $\text{IGDM}_{\text{IAD}}$, 50 for all teachers. For $\text{IGDM}_{\text{AKD}}$, 15 for all teachers. For $\text{IGDM}_{\text{RSLAD}}$, 0.5 for LTD and 0.7 for BDM-AT and IKL-AT. For $\text{IGDM}_{\text{AdaAD}}$, 10 for all teachers.

### B.2.5  ResNet-18 model trained on the SVHN dataset

We set 60 for $\text{IGDM}_{\text{PGD-AT}}$, 30 for $\text{IGDM}_{\text{TRADES}}$, 50 for $\text{IGDM}_{\text{ARD}}$, 40 for $\text{IGDM}_{\text{IAD}}$, 10 for $\text{IGDM}_{\text{AKD}}$, 3 for $\text{IGDM}_{\text{RSLAD}}$, and 9 for $\text{IGDM}_{\text{AdaAD}}$.

### B.2.6  PreActResNet-18 model trained on Tiny-ImageNet dataset

We set 30 for $\text{IGDM}_{\text{PGD-AT}}$, 20 for $\text{IGDM}_{\text{TRADES}}$, 10 for $\text{IGDM}_{\text{ARD}}$, 20 for $\text{IGDM}_{\text{IAD}}$, 10 for $\text{IGDM}_{\text{AKD}}$, 1 for $\text{IGDM}_{\text{RSLAD}}$, and 10 for $\text{IGDM}_{\text{AdaAD}}$.

## C  Main Algorithm

---
**Algorithm 1** Main Algorithm

---
**Input:** Robust teacher model $f_T$, inner loss ($L_{max}$) and outer loss ($L_{AD}$) of base AT or AD method, and batched training dataset $\{(\mathbf{x}, y)\}$ with $n$ batch size.
**Output:** Robust stduent model $f_S$

Randomly initialize $\theta$, the weights of $f_S$
**repeat**
   $\boldsymbol{\delta} = \underset{\|\boldsymbol{\delta'}\|_\infty \leq \epsilon}{\arg\max} \left( L_{max}(\mathbf{x} + \boldsymbol{\delta'}, y) \right)$
   $L_{IGDM}(\mathbf{x}, \boldsymbol{\delta}) = T(\alpha) \cdot D(f_S(\mathbf{x} + \boldsymbol{\delta}) - f_S(\mathbf{x} - \boldsymbol{\delta}),\ f_T(\mathbf{x} + \boldsymbol{\delta}) - f_T(\mathbf{x} - \boldsymbol{\delta}))$
   $L_{min}(\mathbf{x}, y) = L_{AD}(\mathbf{x}, y) + L_{IGDM}(\mathbf{x}, \boldsymbol{\delta})$
   $\theta \leftarrow -\eta \frac{1}{n} \sum_{i=1}^{n} \nabla_\theta L_{min}(\mathbf{x}_i, y_i)$
**until** training converged

---

