# OpenReview forum: "Indirect Gradient Matching for Adversarial Robust Distillation"
_ICLR.cc/2025/Conference — ICLR 2025 Poster_

### Official Review · Reviewer_Pxd3 · 2024-10-29

**Soundness:** 3
**Presentation:** 3
**Contribution:** 3
**Rating:** 5
**Confidence:** 4

**Summary:**

The paper presents an Indirect Gradient Distillation Module (IGDM) for adversarial training via adversarial distillation methods.
It claims to match the input space gradient between teacher and student models with knowledge distillation. The main claim is "indirect" gradient matching/alignment compared to direct gradient alignment, which is available in the literature. Central to the paper is equation 2 (first-order Taylor expansion of adversarially trained models) and equation 8, which is the formulation of IGMD. The paper performs an exhaustive list of experiments. However, there is no indication of making code available for reproducibility is indicated in the paper.

**Strengths:**

The main strength of the paper is the extensive set of experiments on Indirect Gradient Distillation Modules (IGDM) for adversarial training via adversarial distillation methods. The IGDM is use used as an additive loss for adversarial knowledge distillation (Equation 9), which is claimed to be the first Indirect gradient matching compared to direct gradient matching.

**Weaknesses:**

The paper does not make it clear how this way gradient matching is "indirect." From Figure 1 and Figure 2 and the related explanation, it is not clear why this is "indirect" gradient distillation.  There is no comparative picture presented to clarify the comparison/contrast between  Indirect Gradient  and Direct Gradient distillation.

Except for Table 7, It is not explicitly mentioned or clear that they run the other (SOTA) algorithms themselves or that the results were taken from the reported results. If experiments were re-run, other hyperparameter setting descriptions were not given or not referred to which hyperparameter setting were used, e.g., home many epoch os training.

There are some inconsistencies in making results bold for best results in some tables and not in some tables.

**Questions:**

Figure 2a says that the "remainder proportion" of the adversarial robust teachers is shown in Table 1. However, One would wonder, comparing Figure 2a and Table 1, how to make sense of these two different metrics. Figure 2a y-axis is "proportion," Table 1 has clean accuracy and attacked accuracy values. How can one calculate "proportion" from Table 1 to make sense of Figure 2a?

It has not been demonstrated how preparation values 0.012 and 0.016 have arrived. It is not clear if the proposition nose value is the accuracy value. From the equation, it appeared to be a noise term, but making reference to Table 1 makes one feel it is an accuracy term.

Figure 2b also under explains as to how one can relate Equation 2 and Figure 2b.

The paper claims to do an "input gradient" matching Figure 1 and Line 151. However, the gradient matching is done via the discrepancy between student and teacher models model outputs. It is unclear how this method differs from other methods that do gradient distillation.

---

> ### Author Response · Authors · 2024-11-14
> **Reply to Reviewer Pxd3.**
>
> Thank you for your detailed review and constructive feedback. We appreciate your thorough assessment of IGDM and the opportunity to clarify certain points.
>
>
>
> **W1. The paper does not make it clear how this way gradient matching is "indirect." it is not clear why this is "indirect" gradient distillation. There is no comparative picture presented to clarify the comparison/contrast between Indirect Gradient and Direct Gradient distillation.**
>
> **A.** We apologize for any ambiguity in our terminology. Our primary goal is to perform adversarial distillation from a highly robust teacher model, specifically by distilling information from the input gradients, which exhibit critical properties in adversarial training. We use the term “indirect” because our loss design does not calculate the gradient directly; instead, we align gradients through a loss that approximates this effect without explicitly computing the gradient. In Appendix A.4, we provide a comprehensive comparison with methods that perform direct gradient calculation, which further clarifies this distinction.
>
>
> **W2. It is not explicitly mentioned or clear that they run the other (SOTA) algorithms themselves or that the results were taken from the reported results. If experiments were re-run, other hyperparameter setting descriptions were not given or not referred to which hyperparameter setting were used, e.g., home many epoch os training.**
>
> **A.** For all experiments except for Table 7, we ran the baseline algorithms ourselves, strictly following their best hyperparameter settings and training epochs for a fair comparison. We referenced both their official code and the best-performing parameters specified in their papers. Detailed descriptions of experiment settings, including training epochs, are provided in Appendix B.2. For example, for parameters like $\lambda$ in RSLAD and AdaAD, we used $\lambda = 1$ as recommended in their original settings.
>
> **W3. There is no indication of making code available for reproducibility is indicated in the paper.**
>
> **A.** We have already attached our code in the supplementary material. We apologize for not indicating this in the paper.
>
>
> **Q1. Figure 2a says that the "remainder proportion" of the adversarial robust teachers is shown in Table 1**
>
>
> **A.** The caption of Figure 2a is wrong. Sorry for our mistake. We have updated in revision pdf (see blue texts), and the remainder proportion is written in the line number 183-185 of our paper (0.012,0.012,0.016)
>
> **Q2. It has not been demonstrated how preparation values 0.012 and 0.016 have arrived.**
>
> **A.** the proportion values 0.012, 0.012, 0.016 come from equation (2). As you pointed out, Table 1 is about the accuracy of teacher model. sorry again for our mistake, we have updated our paper.
>
> The proportion is calculated as ratio of remainder and total value, i.e., $\frac{\|\|f(\textbf{x} + \boldsymbol{\epsilon}) - f(\textbf{x}) - \left( \frac{\partial f(\textbf{x})}{\partial \textbf{x}} \right)^T \boldsymbol{\epsilon} \|\|}{\|\|f(\textbf{x} + \boldsymbol{\epsilon})\|\|}$.
>
> **Q3. It is unclear how this method differs from other methods that do gradient distillation.**
> **A.** We provide a detailed comparison of other methods that perform gradient distillation within the adversarial distillation context in Appendix A.4.

---

> > ### Author Response · Authors · 2024-11-18
> > **Dear reviewer Pxd3.**
> >
> > Thank you for taking the time to review our submission and provide valuable feedback.
> >
> > We have addressed your comments and uploaded detailed responses. We hope our rebuttal clarifies the points you raised. If you have any additional questions or require further clarifications, we would be more than happy to address them. Thank you again for your time and effort.
> >
> > Best regards,
> >
> > Authors of paper number 5330.

---

> > > ### Comment · Reviewer_Pxd3 · 2024-11-18
> > >
> > > Thanks for the clarification and for acknowledging the mistakes in the original version.

---

> > > > ### Author Response · Authors · 2024-11-18
> > > > **Reply to Reviewer Pxd3.**
> > > >
> > > > Thank you again for taking the time to review our responses. We believe we have addressed the concerns raised, and we hope the clarifications provided help to resolve any remaining uncertainties.
> > > >
> > > > If there are still aspects that are unclear or additional information that could further strengthen our case, we would greatly appreciate your feedback so that we can respond appropriately. If not, we kindly ask you to reconsider the score in light of the additional details and our efforts to improve the work.
> > > >
> > > > We deeply value your time and insights, and we remain open to any further questions or input. Thank you for your thoughtful consideration.

---

> > > > > ### Author Response · Authors · 2024-11-21
> > > > > **Dear Reviewer Pxd3.**
> > > > >
> > > > > We greatly appreciate your recognition of our efforts to address the concerns raised in your initial comments. If there are still any remaining points that require further clarification or areas where additional improvement might strengthen our work, please let us know, and we would be more than happy to provide further responses. As the review deadline is approaching, we wanted to ensure that all concerns are fully addressed in a timely manner.

---

> ### Comment · Reviewer_Pxd3 · 2024-11-29
>
> Yes, Appendix A4 compares the direct gradient and Indirect gradient methods, but not explainaing to the answe why it is Indirect gradient.
>
> I decide to keep my rating as it is.

---

> > ### Author Response · Authors · 2024-11-29
> > **Dear Reviewer Pxd3.**
> >
> > ## 1. Why the term “indirect” gradients was utilized.
> >
> > The term "indirect" is used because, unlike the "direct gradient" approach described in Appendix A4, which explicitly computes gradients using PyTorch’s `autograd` function, our method does not directly extract gradients via `autograd`. Instead, the gradient information is embedded in the output differences, and our loss design leverages this property to align gradients without directly accessing them.
> >
> > This is why we used the term "indirect" to distinguish our approach from directly using the autograd function.
> >
> > We recognize that this reasoning was not explicitly stated in the appendix, and we will revise Appendix A4 to clarify the distinction and explicitly explain why the term "indirect gradient" is appropriate in our context.
> >
> > Thank you again for bringing this to our attention.
> >
> > ## 2. Does our method truly match gradients, despite not calculating gradients directly ?
> >
> >
> >
> > In Table 4 of our main paper, we present quantitative evidence illustrating that integrating our module improves gradient alignment with the teacher. Moreover, the findings in Tables 17 and 18 further support this claim, demonstrating that increasing the weight of IGDM leads to improved gradient alignment: gradient distance (GD) and gradient cosine similarity (GC). Additionally, Equation (4) provides a theoretical basis, indicating that the residual inherently contains gradient information. This is substantiated through both numerical results in Table 4 and theoretical analysis, including a Taylor expansion, which explains how the residual encapsulates gradient information even when derived indirectly through the output.
> >
> > We hope this clarifies your concerns, and please let us know if you have any further questions. Thank you sincerely.

---

### Official Review · Reviewer_yGhV · 2024-11-03

**Soundness:** 3
**Presentation:** 3
**Contribution:** 3
**Rating:** 6
**Confidence:** 5

**Summary:**

This paper proposes a new method named Indirect Gradient Distillation Module (IGDM) for adversarial distillation. Different from the traditional adversarial distillation methods that mainly focus on distilling the logits of the teacher model, this method emphasizes distilling the gradient information of the teacher model. By taking advantage of the local linear characteristics of the adversarially trained model, these gradients are obtained indirectly, thereby significantly enhancing robustness.

**Strengths:**

1. The author obtains the gradients indirectly by taking advantage of the local linear characteristics of the adversarially trained model, thereby significantly enhancing robustness.
2. The modular design of IGDM enables it to be easily integrated with existing adversarial distillation methods.
3. Through a large number of experimental results, the author has verified that IGDM can successfully enhance the robustness of the existing adversarial distillation methods.

**Weaknesses:**

The loss function of the Indirect Gradient Distillation Module (IGDM) proposed in this paper is derived from the first-order Taylor function expansion, which is similar to the situation of making the training loss become low-curvature [in d - f]. In fact, the research works in [e, f] also used similar techniques, but they mainly studied the problems in traditional adversarial training. If the teacher network in this paper is changed to only input natural images, the proposed loss in the paper may degenerate into an evolutionary target of the aforementioned works. Therefore, the author needs to conduct a comparative analysis between their method and the existing works that are also based on the first-order Taylor function expansion, and analyze the differences. Moreover, what will be the result if the regularization loss of the existing works is added to the adversarial distillation target?

The authors utilize the internal loss to generate adversarial perturbations. It would be better to elaborate on its formulaic representation in the main paper. Is the cross-entropy used directly? How much impact do different internal losses have on the results? What would be the effect if the adversarial perturbations are generated using Equation 9?

Some of more recent works, that are about the adversarial distillation, are missing, such as [a-c].

a. Adversarially Robust Distillation by Reducing the Student-Teacher Variance Gap. ECCV 2024

b. PeerAiD: Improving Adversarial Distillation from a Specialized Peer Tutor. CVPR 2024

c. Improving Adversarial Robustness via Information Bottleneck Distillation. NeurIPS 2023

d. Efficient Training of Low-Curvature Neural Networks. NeurIPS 2022.

e. Adversarial Robustness through Local Linearization. NeurIPS 2019.

f. Relating Adversarially Robust Generalization to Flat Minima. ICCV 2021.

**Questions:**

Please check the details in weaknesses.

---

> ### Author Response · Authors · 2024-11-14
> **Reply to Reviewer yGhV.**
>
> Thank you for your detailed review and for highlighting both the strengths and areas for improvement in our work. We appreciate your constructive feedback and the opportunity to clarify certain aspects of our paper.
>
> **W1. Comparative analysis with existing first-order taylor expansion-based methods.**
>
> **A.** Thank you for raising this point. Existing methods in adversarial training [d-f] often aim to achieve low curvature through explicit regularization techniques. In contrast, our approach is distinguished by the **gradient matching** strategy, which is designed to effectively imitate a highly robust teacher model. Instead of directly enforcing low curvature, we leverage the inherent low-curvature property of the teacher model when aligning gradients, using only the output differences.
> Our focus lies in achieving robustness within the adversarial distillation framework, rather than pursuing low curvature explicitly as in existing adversarial training methods. While incorporating additional losses that emphasize low curvature might further enhance performance—and could even be beneficial across other baseline methods—our loss design intentionally shifts focus to a unique perspective that differentiates it from existing techniques.
> We hope this clarifies the novelty and intent of our approach compared to methods based on first-order Taylor function expansions.
>
>
>
> **W2. Clarification on internal loss for generating adversarial perturbations**
>
> **A.** We appreciate your suggestion to provide additional details on the internal loss used to generate adversarial perturbations. As in Appendix B.1, we employed the same inner loss functions as outlined in each original paper: PGD attack on the student model for PGD-AT, ARD, and IAD; TRADES attack on the student model for TRADES; RSLAD inner loss for RSLAD; and AdaAD inner loss for AdaAD. Furthermore, in the methods incorporating IGDM, we followed the inner maximization method specified in the original adversarial training or adversarial distillation method.
> For formulaic representation as follows,
>
> PGD attack :  $\boldsymbol{\delta} = \text{arg} \max_{ \|\| \boldsymbol{\delta'} \|\|_p \leq \epsilon} \text{CE} (f_S  (\mathbf{x}+\boldsymbol{\delta'}), y)$
>
> TRADES attack :  $\boldsymbol{\delta} = \text{arg} \max_{ \|\| \boldsymbol{\delta'} \|\|_p \leq \epsilon} \text{KL} (f_S  (\mathbf{x}+\boldsymbol{\delta'}), f_S  (\mathbf{x}))$
>
> RSLAD attack : $\boldsymbol{\delta} = \text{arg} \max_{ \|\| \boldsymbol{\delta'} \|\|_p \leq \epsilon} \text{KL} (f_S  (\mathbf{x}+\boldsymbol{\delta'}), f_T  (\mathbf{x}))$
>
> AdaAD attack : $\boldsymbol{\delta} = \text{arg} \max_{ \|\| \boldsymbol{\delta'} \|\|_p \leq \epsilon} \text{KL} (f_S  (\mathbf{x}+\boldsymbol{\delta'}), f_T  (\mathbf{x} +\boldsymbol{\delta'}))$
>
> Here, $\text{CE}$ represents cross-entropy loss, $\text{KL}$ represents KL-divergence loss.
>
> **Q1. What would be the effect if the adversarial perturbations are generated using Equation 9?**
>
> **A.** We also experimented with using Equation 9 to generate adversarial examples; however, this did not lead to significant performance improvements. This outcome may be due to the necessity of additional hyperparameter tuning and adjustments to effectively utilize Equation 9 for attacks. Designing a more effective inner maximization process remains an avenue for future work.
>
>
> **W3. Missing related works**
>
> **A.** Thank you for pointing out recent related works on adversarial distillation. We apologize for the oversight and will update the paper to include citations. Here is our perspective on each paper:
> - *Adversarially Robust Distillation by Reducing the Student-Teacher Variance Gap (ECCV 2024)* : This is a concurrent paper that was published shortly before we completed our manuscript, which is why it was not included initially. Currently, there is no official code available, so we are unable to reproduce their results for direct comparison in our main paper.
>
> - *PeerAiD: Improving Adversarial Distillation from a Specialized Peer Tutor (CVPR 2024)* : This approach differs slightly from traditional adversarial distillation in that it uses a peer model instead of a teacher model, training both the peer model and the student together. Due to this setup, PeerAiD does not utilize a strong, robust teacher, which results in lower reported performance metrics. Because of these fundamental differences in the experimental setup, we decided not to include PeerAiD in our main table for direct comparison.
>
> - *Improving Adversarial Robustness via Information Bottleneck Distillation (Neurips 2023)* : First, this paper has not been widely cited by other adversarial distillation studies. We believe this may be due to limited code availability, which affects reproducibility. Additionally, its absolute performance are lower than those of more recent work, such as AdaAD, so we decided not to use it as a baseline.

---

> > ### Author Response · Authors · 2024-11-18
> > **Dear reviewer yGhV.**
> >
> > Thank you for taking the time to review our submission and provide valuable feedback.
> >
> > We have addressed your comments and uploaded detailed responses.
> > We hope our rebuttal clarifies the points you raised.
> > If you have any additional questions or require further clarifications, we would be more than happy to address them.
> > Thank you again for your time and effort.
> >
> >
> > Best regards,
> >
> > Authors of paper number 5330.

---

> > > ### Author Response · Authors · 2024-11-21
> > > **Dear reviewer yGhV.**
> > >
> > > We add more experiments with our explanation of the difference between from [d-f] and our proposed method.
> > >
> > > First, [f] analyzes a strong correlation between loss flatness and good robust generalization. In Figure 1 of [f], it is evident that as the model becomes flatter, robustness increases as well.
> > > This study investigates the flatness of weights and analyzes the relationship between weight flatness and robustness, rather than aiming to enhance the performance of adversarial training (AT).
> > >
> > > In the case of [d] and [e], we conducted experiments by adding a loss term targeting flatness, specifically aiming to constrain loss curvature, in response to the reviewer’s question.
> > > We referred to the official code.
> > >  (Reference: [LCNN GitHub Repository](https://github.com/kylematoba/lcnn)).
> > > These experiments were conducted using the BDM teacher on the CIFAR-100 dataset.
> > >
> > > | Method                    | Clean  | AA    |
> > > |---------------------------|--------|-------|
> > > | ARD                      | 61.51  | 24.77 |
> > > | ARD + Low curvature      | 63.27  | 22.74 |
> > > |---------------------------|--------|-------|
> > > | IAD                      | 59.92  | 25.15 |
> > > | IAD + Low curvature      | 61.42  | 24.31 |
> > > |---------------------------|--------|-------|
> > > | AKD                      | 60.27  | 25.09 |
> > > | AKD + Low curvature      | 55.27  | 20.61 |
> > > |---------------------------|--------|-------|
> > > | RSLAD                    | 60.22  | 27.96 |
> > > | RSLAD + Low curvature    | 61.92  | 25.61 |
> > > |---------------------------|--------|-------|
> > > | AdaAD                    | 64.43  | 28.06 |
> > > | AdaAD + Low curvature    | 65.00  | 26.48 |
> > > |---------------------------|--------|-------|
> > > | AdaIAD                   | 64.13  | 27.82 |
> > > | AdaIAD + Low curvature   | 64.80  | 26.08 |
> > > |---------------------------|--------|-------|
> > > | AdaAD + IGDM                   | 64.44  | 30.32 |
> > > | AdaAD + IGDM + Low curvature    | 65.03  | 28.02 |
> > >
> > >
> > >
> > > We also conducted hyperparmeter searching for low curvature loss. In this experiment, we utilize ARD since ARD is the most basic adversarial distillation method.
> > >
> > >
> > > | Method                    | Clean  | AA    |
> > > |---------------------------|--------|-------|
> > > | ARD + Low curvature ($\lambda = 0.01$) | 61.81 | 24.34 |
> > >  | ARD + Low curvature ($\lambda = 0.02$) | 62.08 | 24.04 |
> > >  | ARD + Low curvature ($\lambda = 0.05$) | 62.81 | 23.14 |
> > >  | ARD + Low curvature ($\lambda = 0.1$) | 63.27 | 22.74 |
> > >
> > > Here, $\lambda$ denotes the hyperparameter for the Low Curvature Loss.
> > >
> > >
> > >
> > >
> > > As shown in the results, using the low curvature loss slightly increases clean accuracy but decreases robust accuracy (AA accuracy). This aligns with the findings of the original paper [e], where the caption of Table 2 states: *”Adversarial training performs the best overall, however sacrifices clean accuracy. LCNN+GradReg models perform similarly but without significant loss of clean accuracy."* In the numerical results of Table 2 in paper [e], we can also see that adding a loss term that promotes low curvature leads to a decrease in robustness compared to basic PGD-AT, while clean accuracy slightly increases.
> > > Originally, paper [e] argues that flatness can be used to **efficiently train a robust model without adversarial generation**, rather than achieving **additional robustness improvement** through flatness.
> > >
> > > Therefore, even if we additionally use a loss that constrains low curvature in the adversarial distillation framework, the performance improvement is minimal, as shown in our experiment. This is because we are distilling robustness from an already robust teacher model that possesses flatness, so enforcing flatness explicitly when training the student does not seem to increase performance.
> > > Our paper does not explicitly specify that the student model should have flatness. Instead, by utilizing the flatness that arises from adversarial training, we derive that the output difference includes gradient information, and by distilling the gradient, we explained that the point-wise matching between the teacher and student increases.
> > >
> > > To summarize, while existing methods [d-f] incorporate a loss term to enforce flatness and explore the relationship between flatness and robustness, our approach does not include a loss term explicitly targeting flatness. Instead, the key difference lies in leveraging gradient matching through output differences to effectively mimic the teacher model. This allows us to focus on enhancing the student model's performance without directly enforcing flatness constraints.
> > >
> > >
> > > We truly appreciate your valuable feedback and have addressed the points raised in our response. If there are any additional questions or concerns, we would be glad to further clarify. Given the upcoming deadline for the review phase, we would like to ensure that all remaining queries are resolved in a timely manner.

---

> > > > ### Author Response · Authors · 2024-11-25
> > > > **Dear reviewer yGhV.**
> > > >
> > > > As the discussion period is ending soon, we wanted to remind you about our responses to your review.
> > > >
> > > > We have carefully addressed your comments and hope they resolve your concerns. If you have any further questions, please let us know before the deadline.
> > > >
> > > > Thank you for your time and feedback.

---

> > > > > ### Author Response · Authors · 2024-11-28
> > > > > **Dear reviewer yGhV.**
> > > > >
> > > > > As the discussion period is coming to a close, we wanted to make a final follow-up regarding your review.
> > > > >
> > > > > We have thoroughly addressed your comments and sincerely hope our responses have resolved your concerns. If you have any further questions or feedback, we would be grateful if you could share them with us before the discussion deadline.
> > > > >
> > > > > Thank you once again for your time and valuable insights throughout this process.

---

### Official Review · Reviewer_avUv · 2024-11-03

**Soundness:** 3
**Presentation:** 3
**Contribution:** 3
**Rating:** 8
**Confidence:** 4

**Summary:**

The paper focuses on improving the adversarial robustness of DNNs through adversarial distillation. Specifically, it addresses the challenge that smaller models, which are preferred for their computational efficiency, lag behind larger models in terms of robustness against adversarial attacks. The authors propose the Indirect Gradient Distillation Module (IGDM), which aims to transfer the input gradient knowledge from a robust teacher model to a student model, indirectly matching the student’s input gradient with the teacher's.  The experimental results demonstrate the effectiveness of the proposed approach in significantly improving adversarial robustness.



Based on the detailed response by the authors, I decided to improve my rating!

**Strengths:**

1. The authors propose a novel distillation module called the Indirect Gradient Distillation Module (IGDM) and provide a theoretical foundation for why gradient matching through output differences works, leveraging the local linearity of adversarially trained models. But, I have some doubts about the theoretical assumptions here. The fragility of neural networks is attributed to the linear properties of neural networks[1], and the paper assumes this local linear property to solve the gradient, which is contrary to the conclusions of previous related research. So, does this method really simulate the gradient of the network?

2. Extensive experiments across different datasets, student models, and teacher models validate the effectiveness of IGDM in improving robustness against various attack scenarios.

[1] Ian J Goodfellow, Jonathon Shlens, and Christian Szegedy. Explaining and harnessing adversarial examples. ICLR, 2015.

**Weaknesses:**

1.  Although the paper did a lot of experiments, I think it is not fair enough. For example, the teacher model used in the paper is usually better than the teacher models of other methods. So if other methods also use the same teacher model, can they also achieve good robustness?

2. Line 160 of the paper mentions that using gradients directly is difficult, which is important for the motivation of the paper, so it needs to be reflected in the main context rather than being placed in the Appendix.

**Questions:**

1. I am confused about Figure 1(a) in the paper. In the distillation architecture, the teacher model is already trained, while the student model is not. In the feature space, the manifolds of the models are different. Even if the directions of the gradients are the same, the manifold spaces are inconsistent, and the paths they take should also be different. Therefore, the adversarial samples of the teacher model and the student model cannot be consistent. One way to prove this is to use the adversarial samples generated by the teacher model and the student model to attack another model to see if the attack effects are consistent.

2  In Figure 2, in adversarial training and adversarial distillation, why is it that in the early stages of training, when the student model should not be robust yet, why does the remainder proportion of the model not increase?

---

> ### Author Response · Authors · 2024-11-14
> **Reply to Reviewer avUv.**
>
> Thank you for your thoughtful review and for raising insightful questions about our work. We appreciate your recognition of IGDM’s novel approach and its effectiveness in improving adversarial robustness through extensive experiments.
>
> **Strength-Q1. Addressing concerns about theoretical assumptions, does this method really simulate the gradient of the network?**
>
> **A.** We acknowledge your concern regarding the assumption of local linearity in our gradient distillation approach. While previous studies [1] attribute the fragility of neural networks to their linear properties, we want to clarify that [1] refers to linear networks, which focus on the *network weights*. Our observation, however, is based on the linear property *with respect to the inputs*, not the network weights. The linear property with respect to inputs is observed as shown in Figure 2-(b).
> To address your question, we also numerically verify that our method simulates the gradients of the network, as shown in Table 4. Specifically, we calculate the gradients with respect to the input for both the teacher and student networks. We then observe that the gradient distance and gradient cosine similarity become closer, demonstrating that our method effectively simulates the gradients with respect to the inputs of the network.
>
>
>
> **W1. If other methods also use the same teacher model, can they also achieve good robustness?**
>
> **A.**  We respectfully disagree with the assertion that our comparisons are unfair. To clarify, in all of our experiments, we used the **same teacher model** consistently across different methods within each experimental setting.
> More detailed reasons are as follows:
> First, as shown in Table 1, for CIFAR-10 and CIFAR-100, we used publicly available models from RobustBench, ensuring accessibility to all researchers. For SVHN and Tiny-ImageNet, since prior works like RSLAD and AdaAD do not report experiments, we trained teacher models ourselves and plan to make these models publicly available. Additionally, the CIFAR-100 LTD model in Table 1 is the exact same teacher model used in AdaAD, ensuring consistency across methods. Thus, our choice of teacher models was not intended to artificially boost IGDM’s performance by cherry-picking but rather to ensure a fair comparison based on widely recognized or directly comparable models.
>
> Second, as shown in Figure 4, we used teachers which are publicly available models from RobustBench. Note that all the baselines utilize the **same** teacher to ensure fair comparison. While a stronger teacher generally improves student performance, we observe cases—such as in Table 3 vs. Table 13 on CIFAR-10—where a lower-performing teacher yields higher distillation outcomes. This suggests that teacher robustness does not always directly correlate with student performance. We find this observation particularly interesting, as it opens up new avenues for future work to explore how teacher model robustness impacts student performance in adversarial distillation.
>
>
> **W2. Placement of gradient discussion in the main text.**
>
> **A.** We appreciate the suggestion to bring the discussion of direct gradients matching into the main body of the paper. We agree that this point is indeed important for the motivation of the paper and would ideally be included in the main context. However, as it is tangential to the specific details of the IGDM module itself, and due to page limitations, we place it in the Appendix.
>
> **Q1. Inconsistency of manifold spaces between teacher and student.**
>
> **A.**  As you pointed out, the manifolds of the teacher and student models are different, which means their gradient directions may not perfectly align, and the adversarial samples they generate will not be identical. We did not claim that the adversarial samples of the teacher and student models are the same. Rather, we use adversarial samples generated by the student model to match its gradients with those of the teacher model. This gradient matching enables the student model to better imitate the teacher, as shown in Equation (10). By aligning the gradients, we achieve point-wise alignment between the student and teacher, as demonstrated in Table 5. Figure 1(a) provides a conceptual illustration of this point-wise alignment, which is also numerically verified in Table 5.
>
> **Q2. Why does the remainder proportion of the model not increase in the early stages of training?**
>
> In Figure 2, since we plotted the remainder proportion per epoch, it appears lower in the early stages of training. If we were to plot it per step, the remainder proportion would be higher in the very early stage. Our experimental findings indicate that adversarial training results in a lower remainder proportion compared to natural training, even at the beginning of training. We believe this lower remainder proportion is a characteristic of adversarial training, which we have observed experimentally.

---

> > ### Author Response · Authors · 2024-11-18
> > **Dear reviewer avUv.**
> >
> > Thank you for taking the time to review our submission and provide valuable feedback.
> >
> > We have addressed your comments and uploaded detailed responses.
> > We hope our rebuttal clarifies the points you raised.
> > If you have any additional questions or require further clarifications, we would be more than happy to address them.
> > Thank you again for your time and effort.
> >
> >
> > Best regards,
> >
> > Authors of paper number 5330.

---

> > ### Public Comment · ~Xinyu_Zhang15 · 2025-03-13
> > **Questions about the estimation of remainder**
> >
> > Dear Author,
> > I have read the method for calculating the remainder in your paper, but encountered difficulties in reproducing the results. When directly computing the product of \nabla_x f(x) and \delta using PyTorch, there is a significant discrepancy between this value and the actual difference \delta(x+\delta) - f(x), and the ratio becames very large. I would appreciate your guidance on resolving this issue.

---

> > > ### Public Comment · ~Xinyu_Zhang15 · 2025-03-13
> > > **Question solved**
> > >
> > > I am very grateful to the author for solving my problem in detail!

---

> ### Comment · Reviewer_avUv · 2024-11-18
> **reply to the response**
>
> Thanks for the author's response, which solved some of my concerns. I still have a few questions.
>
> 1. For the teacher model, I checked the results of the original RSLAD paper, and its teacher model (WideResNet-34-10) has an accuracy of 26% under AA attack. The student model can achieve an accuracy of 26.7%, while in this manuscript, a more powerful teacher model (~38%) is used. But the defense of the student model is not improved; it is still 26.7 in Table 2.
>
> 2. As Reviewer Pxd3 mentioned, I don't think the author's method fits the gradient. I think the proposed method still fits the model's output distribution in the form of residuals.

---

> ### Author Response · Authors · 2024-11-18
> **Reply to Reviewer avUv.**
>
> Thank you for your thoughtful questions and for bringing up this important point.
>
> **Q1-1. I checked the results of the original RSLAD paper, and its teacher model (WideResNet-34-10) has an accuracy of 26% under AA attack.**
>
> **A.** We would like to clarify a point regarding the teacher model used in the original RSLAD paper. On page 5 of the RSLAD paper, it explicitly states:
>
> * "For CIFAR-100, we use the WideResNet-70-16 model provided by Gowal et al." in the section 4.1.
>
> Additionally, we verified that the official code for RSLAD also utilizes the WideResNet-70-16 teacher model for CIFAR-100.
> (Reference: [RSLAD GitHub Repository](https://github.com/zibojia/RSLAD/blob/c40e04f6c4edcd87431d0250b168d503b1293745/resnet18_rslad_cifar100.py#L43C1-L43C70)).
>
> The Auto Attack (AA) performance of the WideResNet-70-16 teacher model on CIFAR-100, as reported in the original paper and verified by us, is **30.03%**, not **26%**. We hope this clarifies any confusion regarding the teacher model and its performance in the original RSLAD experiments.
>
>
>
>
> **Q1-2. the defense of the student model has not improved; it is still 26.7**
>
> **A.**
> Regarding the teacher model, we refer to the RSLAD paper (Chapter 4.4, "How to Choose a Good Teacher?"), which states:
>
> * ”We find that the student’s robustness does not increase monotonically with that of the teacher’s; instead, it first rises then drops. We call this phenomenon robust saturation.”
>
> This phenomenon is clearly illustrated in Figure 4 of the RSLAD paper on CIFAR-10, where the WideResNet-34-10 teacher model with an AutoAttack robustness of 53.08% achieves the highest student robustness (\~52%). Interestingly, using a stronger teacher model, such as WideResNet-76-10 with an AutoAttack robustness of 57.20%, results in lower student robustness (\~51%).
>
> Therefore, the argument that our RSLAD results should improve from 26.7% simply because we used a stronger teacher does not hold, as the original RSLAD findings already establish this inconsistency. The phenomenon of **robust saturation** highlights that a more robust teacher does not necessarily lead to a more robust student, as discussed in the original RSLAD paper.
>
> In contrast, our method clearly demonstrates performance improvement over RSLAD under the same teacher setting. This is a key distinction: while RSLAD fails to fully leverage the increased robustness of a stronger teacher, our approach successfully overcomes this limitation. For instance, using the BDM-AT teacher (AA: 38.83%), RSLAD achieves an AA of 26.76%, whereas RSLAD+IGDM achieves a significantly improved AA of 29.78%.
>
>
> **Q2. I don't think the author's method fits the gradient.**
>
> **A.**
> In Table 4, we provided quantitative evidence demonstrating that incorporating our module enhances the gradient matching with the teacher. Additionally, the results in Tables 17 and 18 further validate this point, which shows that as the weight of IGDM increases, the degree of gradient matching (GD, GC) also improves. Furthermore, Equation (4) offers a mathematical explanation, showing that the residual effectively contains gradient information. We supported this with both numerical data in Table 4 and theoretical analysis, including clarification through Taylor expansion that the residual captures gradient information, even if approximated through the output.
>
> We would appreciate any further insights from the reviewer if there are still concerns about the validity of these results, despite the clear evidence presented. Reviewer pxd3 has kindly acknowledged that all concerns have been addressed. If there are additional points for discussion, we would be grateful to hear them. Thank you for your thoughtful consideration.

---

> ### Comment · Reviewer_avUv · 2024-11-19
> **Reply to the response**
>
> Thanks for the author's response.
>
> Thanks to the author for pointing out the information about the teacher model, and I apologize for that. Actually, I am confused as to why the student model did not improve with a more robust teacher model, even though RLSAD proposed robust saturation for this phenomenon, but there is no similar phenomenon in other distillation methods.
> In the submitted manuscript, you just used a 28x10 WideResNet, IDGM got excellent results, but RLSAD did not improve.

---

> > ### Author Response · Authors · 2024-11-19
> > **Reply to Reviewer avUv.(1/2)**
> >
> > We sincerely thank the reviewer for replying to our answer again.
> > Thank you for raising this interesting question. To address your inquiry, we have organized our response into two parts: (1) experimental observations based on rigorous testing, and (2) speculative reflections on the phenomenon of robust saturation. We would like to emphasize that the discussions in part **(2) are not directly related to the core contributions of our manuscript submitted to ICLR.** These points are shared to provide additional context and address the curiosity expressed in your comment.
> >
> > ---
> >
> > ### (1). **Experimental observations based on rigorous testing**
> >
> > ### (1) - a. Why robust saturation is less reported in other adversarial distillation methods
> >
> > We believe the lack of reports on robust saturation in other adversarial distillation (AD) methods stems from limited experimentation with diverse teacher models in most existing works. Many AD studies focus on evaluating a single-teacher model and do not thoroughly explore the dynamics of teacher-student performance across varying teacher robustness levels.
> >
> > From our experiments, robust saturation is not unique to RSLAD but occurs in multiple AD methods when tested with diverse teacher models. As shown in our paper (Figure 4), using a moderately robust teacher generally improves the student's robustness as expected. However, this improvement is not guaranteed when a highly robust teacher is used. We observed this behavior in methods like ARD, AKD, IAD, RSLAD, AdaAD, and our proposed IGDM. This indicates that robust saturation is a general phenomenon across AD methods.
> >
> > For example, in Table 3 (CIFAR-10 with LTD teacher, AA: 56.94) and Table 13 (CIFAR-10 with IKL-AT teacher, AA: 67.73), we observe that the LTD teacher often results in better student performance, even though the IKL-AT teacher has higher robustness.
> > In CIFAR-100, in the table 2, The AA accuracy of BDM and LTD teacher is **38.87%** and **30.57%**. We also observe that the LTD teacher often results in better student performance, even though the BDM teacher has higher robustness.
> > Specifically, the AA accuracy of baseline models is as follows.
> > The AA accuracy of each baseline student model is as follows.
> >
> > + ARD : 24.77 ( with 38.87 BDM teacher) vs  25.74 (with 30.57 LTD teacher)
> >
> > + IAD : 25.15 ( with 38.87 BDM teacher)  vs 26.19 (with 30.57 LTD teacher)
> >
> > + AKD : 25.09 ( with 38.87 BDM teacher) vs 25.37 (with 30.57 LTD teacher)
> >
> >
> > + RSLAD :  27.96 (with 38.87 BDM teacher) vs 26.94 (with 30.57 LTD teacher)
> >
> > + AdaAD :  28.06 (with 38.87 BDM teacher) vs  27.81 (with 30.57 LTD teacher)
> >
> > + AdaIAD : 27.82 (with 38.87 BDM teacher) vs  27.83 (with 30.57 LTD teacher)
> >
> >
> >
> > This demonstrates that using a stronger teacher does not always yield a more robust student, corroborating the robust saturation phenomenon described in RSLAD.

---

> > > ### Author Response · Authors · 2024-11-19
> > > **Reply to Reviewer avUv.(2/2)**
> > >
> > > ### (2). **Speculative reflections on robust saturation**
> > >
> > > ### (2) - a. Observed trends in robust saturation
> > >
> > > From our experiments, we identified the following trends:
> > > - **Teacher-student performance consistency:** From our experiments, we observed a consistent trend in teacher-student performance relationships. If a highly robust teacher results in lower student performance compared to a less robust teacher when using a SOTA AD method, this trend generally persists across other AD methods. This indicates that both the teacher-student pairing and the choice of AD method influence the results.
> > > - **Dataset complexity matters:** Robust saturation is more frequently observed in simpler datasets like CIFAR-10 compared to more complex datasets like CIFAR-100. For more complex datasets, the phenomenon appears less pronounced.
> > >
> > > ### (2) - b. Hypotheses about robust saturation
> > >
> > > While not the primary focus of our manuscript, we hypothesize the following potential causes for robust saturation based on our observations:
> > >
> > > - **Model capacity limitations:** Smaller models like ResNet-18 may have capacity ceilings that prevent the student from fully leveraging the robustness of highly robust teachers. Beyond a certain threshold, increasing teacher robustness may not improve the student's robustness due to these capacity limitations. The relation of  model capacity and the robust performance can be found in [1] (*”To reliably withstand strong adversarial attacks, networks require a larger capacity than for correctly classifying benign examples only.”*, in page 2 in [1]).
> > >
> > >
> > > [1] Madry, Aleksander. "Towards deep learning models resistant to adversarial attacks." arXiv preprint arXiv:1706.06083 (2017).
> > >
> > > - **Robust multi-view features:** Similar to knowledge distillation paper [2], where KD can be viewed as an ensemble of single-view features to learn multi-view features, adversarial distillation may involve robust multi-view features. Some of features from robust teacher may be easier for smaller models to learn, while others may be inaccessible. This could explain why robust saturation or performance reversals occur with stronger teachers, which have hard-to-learn robust features.
> > >
> > > [2] Allen-Zhu, Zeyuan, and Yuanzhi Li. "Towards understanding ensemble, knowledge distillation and self-distillation in deep learning." arXiv preprint arXiv:2012.09816 (2020).
> > >
> > > ### 3. **Disclaimer on these insights**
> > >
> > > We reiterate that the above hypotheses are speculative and not part of the main contributions of our manuscript. They are shared here to address the reviewer's curiosity about robust saturation. We hope this discussion provides additional insight and encourages further exploration of this phenomenon in adversarial distillation. Thank you again for your thoughtful question.

---

> > > > ### Comment · Reviewer_avUv · 2024-11-20
> > > > **Reply to the response**
> > > >
> > > > Thanks to the author for the detailed reply, which solved my concerns. I decide to improve my rating !

---

> > > > > ### Author Response · Authors · 2024-11-20
> > > > > **Reply to Reviewer avUv.**
> > > > >
> > > > > Thank you for taking the time to consider our response and for addressing your concerns. We sincerely appreciate your constructive feedback throughout this process, and we’re glad our explanations were helpful. Thank you for your thoughtful review.

---

### Official Review · Reviewer_ECU4 · 2024-11-04

**Soundness:** 3
**Presentation:** 3
**Contribution:** 3
**Rating:** 8
**Confidence:** 3

**Summary:**

This paper aims to improve the adversarial robustness of lightweight or small models using adversarial distillation. Unlike directly using the teacher's logits as a guide, the authors focus on distilling the input gradients of the teacher model to achieve point-wise alignment between the teacher and student. The proposed IGDM can integrate with other AD methods and has demonstrated consistent performance improvements across different datasets.

**Strengths:**

1. Aligning the input gradients of the teacher and student is an insightful and interesting approach.

2. IGDM is a simple and effective method that can be easily applied as a regularization term to other AD methods, achieving consistent performance improvements.

3. Comprehensive experimental results and ablation studies demonstrate the effectiveness of IGDM.

4. The paper is well-written and easy to follow.

**Weaknesses:**

Overall, this paper is logically coherent and well-argued. The effectiveness of IGDM is based on the assumption of the locally linear property of adversarial robust models, and preliminary experiments in Section 3.1 support this assumption. However, the data and models used in the experiments are relatively simple. Validating the effectiveness of IGDM on more general models, such as ViT or CLIP, and on more complex tasks, such as detection and segmentation, could provide more universal insights for future research.

**Questions:**

See Weaknesses

---

> ### Author Response · Authors · 2024-11-14
> **Reply to Reviewer ECU4.**
>
> Thank you for your positive evaluation and thoughtful feedback on our work. We appreciate your recognition of IGDM’s innovative approach in aligning input gradients between teacher and student models, as well as your comments on the clarity and accessibility of the paper. Your insights will be invaluable as we continue to improve and broaden the scope of IGDM.
>
> **W1. Validation on more complex models and tasks could provide more universal insights for future research**
>
> **A.** We acknowledge your suggestion to evaluate IGDM on more complex models, such as ViT or CLIP, and on more advanced tasks like detection and segmentation. We agree that validation on more diverse architectures and complex tasks would further strengthen IGDM’s general applicability. Expanding our experiments in these directions is a priority for future work, as it will provide additional insights into IGDM’s robustness and scalability.

---

### Author Response · Authors · 2024-12-02
**Global Response**

Dear Reviewers,

We sincerely thank you for your thoughtful questions and constructive feedback. Your insights have been instrumental in improving the quality of our work. We have carefully addressed all the concerns raised in our rebuttal and incorporated revisions in the manuscript.
First, we deeply appreciate your recognition of the following strengths in our work:

- The extensive experiments demonstrating the effectiveness of IGDM across diverse settings.

- Our method's modular design allows for seamless integration into existing adversarial distillation (AD) frameworks.

Additionally, we are grateful for the opportunity to clarify and address several key concerns raised in the reviews:

1. **Fairness of Experimental Settings** (Reviewer **avUv**):
   One concern was that our experiments employed more advanced and robust teacher models compared to prior AD methods. To address this, we showed that our experiments were conducted with diverse teacher models, including some used in prior works. Furthermore, we provided a detailed analysis of the robust saturation phenomenon proposed in RSLAD, offering additional experimental insights that resolve this concern. As a result, we successfully addressed the concern raised by reviewer avUv. The reviewer also responded with gratitude, acknowledging that their concern was resolved.

2. **Comparison with Low-Curvature Losses in AT** (Reviewer **yGhV**):
   Another concern was the comparison between IGDM and low-curvature-related losses in prior adversarial training methods. Our additional experiments during the rebuttal period showed that incorporating low-curvature losses into existing AD methods often degraded robust performance (in reply to yGhV).  Furthermore, we also clarified that IGDM focuses on distilling gradient information from a well-trained teacher rather than merely reducing curvature. This distinction provides a clearer understanding of the complementary role of IGDM.


3. **"Indirect" Gradient Matching Explanation** (Reviewer **Pxd3**, Reviewer **avUv**):
 We addressed concerns regarding the indirect nature of gradient matching by providing theoretical insights through Taylor expansion, illustrating how the linear properties enable logit differences to effectively encode gradient information. Furthermore, we empirically demonstrated that logit differences indeed encapsulate gradient information by measuring gradient alignment using metrics, GC and GD, directly computed via `autograd`.



We have also improved the overall clarity of the manuscript and corrected all identified typos to ensure the paper is as clear and polished as possible. All modifications are highlighted in BLUE in the revised manuscript.
We sincerely hope that these revisions address the reviewers' concerns and enhance the overall quality of the paper. Thank you once again for your valuable feedback and for engaging deeply with our work.

Best regards,

The Authors

---

### Meta-Review · Area_Chair_67mq · 2024-12-23

**Metareview:**

This paper investigates adversarial distillation, focusing on small student models. Compared to vanilla distillation strategies, a key enhancement here is to indirectly match the input gradients between a robust teacher model and a student model. The empirical results demonstrate consistent improvements over baseline methods.

Overall, the reviewers found the paper well-written and easy to follow, and they appreciated its novelty and effectiveness. But meanwhile, several major concerns are raised, including 1) the term "Indirect Gradient Matching" (especially regarding the word "indirect") is somewhat unclear and confusing; 2) missing discussions about prior related works that enforce low curvature; and 3) some experimental setups, as well as tables and figures, require further explanation and clarification.

During the rebuttal and the discussion stage, the authors actively provided additional experiments and clarifications to address these concerns. As a result, three reviewers positively rate the quality of this paper. Reviewer Pxd3 remained slightly negative about this submission, but the AC found no concerning points remaining from their feedback. Therefore, the final decision is acceptance.

The authors must integrate the additional experiments, clarifications, and discussions from the rebuttal into the camera-ready version and correct any typos or mistakes pointed out by the reviewers.

**Additional Comments On Reviewer Discussion:**

The reviewers' major concerns are listed in the meta-review. During the rebuttal and discussion stages, the authors sufficiently addressed all these concerns.

With no major issues remaining, and considering the novelty and effectiveness of the proposed method, the AC recommends accepting this submission.

---

### Decision · Program_Chairs · 2025-01-22

Accept (Poster)